*Proc. R. Soc. B* **287**: 20200763.

biomechanics, ecology, evolution

contemporary evolution, domestication, urban ecology, morphometrics, developmental bias, Canidae

**Author for correspondence:**
K. J. Parsons
e-mail: kevin.parsons@glasgow.ac.uk

# Skull morphology diverges between urban and rural populations of red foxes mirroring patterns of domestication and macroevolution

K. J. Parsons[1], Anders Rigg[1], A. J. Conith[2], A. C. Kitchener[3,4], S. Harris[5] and Haoyu Zhu[1]

[1]Institute of Biodiversity, Animal Health, and Comparative Medicine, University of Glasgow, Glasgow G12 8QQ, UK
[2]Department of Biology, University of Massachusetts, Amherst, MA 01003, USA
[3]Department of Natural Sciences, National Museums Scotland, Chambers Street, Edinburgh EH1 1JF, UK
[4]Institute of Geography, School of Geosciences, University of Edinburgh, Drummond Street, Edinburgh EH8 9XP, UK
[5]School of Biological Sciences, University of Bristol, Bristol BS8 1TQ, UK

KJP, 0000-0003-3355-3587; AJC, 0000-0001-9357-6620

Human activity is drastically altering the habitat use of natural populations. This has been documented as a driver of phenotypic divergence in a number of wild animal populations. Here, we show that urban and rural populations of red foxes (*Vulpes vulpes*) from London and surrounding boroughs are divergent in skull traits. These changes are primarily found to be involved with snout length, with urban individuals tending to have shorter and wider muzzles relative to rural individuals, smaller braincases and reduced sexual dimorphism. Changes were widespread and related to muscle attachment sites and thus are likely driven by differing biomechanical demands of feeding or cognition between habitats. Through extensive sampling of the genus *Vulpes*, we found no support for phylogenetic effects on skull morphology, but patterns of divergence found between urban and rural habitats in *V. vulpes* quantitatively aligned with macroevolutionary divergence between species. The patterns of skull divergence between urban and rural habitats matched the description of morphological changes that can occur during domestication. Specifically, urban populations of foxes show variation consistent with 'domestication syndrome'. Therefore, we suggest that occurrences of phenotypic divergence in relation to human activity, while interesting themselves, also have the potential to inform us of the conditions and mechanisms that could initiate domestication. Finally, this also suggests that patterns of domestication may be developmentally biased towards larger patterns of interspecific divergence.

## 1. Introduction

Human activity often results in rapid and extreme environmental variation with biologists becoming increasingly interested in predicting how populations will respond to such novel changes [1]. The capacity for developmental changes in the expression of adaptive phenotypes (i.e. evolvability) could be key for allowing animals to persist in anthropogenic environments [2]. For example, cities may represent extremely novel environments with alterations of food availability, predation, spatial structure, lighting period, community structure, as well as auditory and visual cues. While there is ample evidence of phenotypic divergence between rural and urban populations of animals, we still know little about how selection could be differentially operating between these environments across a suite of functionally salient traits, and across replicate habitat gradients in other cities

[3]. Thus, while phenotypic divergence occurs between urban and rural habitats, it could be that some changes are due to variation in cognitive demands, while others are due to variation in biomechanical demands. Indeed, prior research suggests that a larger brain size can facilitate the innovative behaviours that could allow animals to persist in urban habitats, including the exploitation of novel foods, or the evasion of novel threats [1–3, but see 4]. Characterizing the morphological changes during a transition from a rural to urban environment could indicate how wild populations specifically cope with living in close proximity to humans but may also provide insights into how domestication is initiated.

Domestication results in a number of stereotypical changes known as 'domestication syndrome' [5–9]. Domestication syndrome refers to the suite of phenotypic changes that are known to occur in response to the domestication process. For example, domestication leads to stereotypical changes across species toward more docile behaviour, coat colour changes, reduced total brain size, reductions in tooth size, prolongations of juvenile behaviour, and changes in craniofacial traits, including a shortened skull morphology. Such changes are well documented among several taxa, with foxes and domestic dogs being particularly well studied [9–12]. Domestic dogs underwent extensive morphological change to form today's modern dog breeds with especially notable changes in the skull that are primarily characterized by snout lengthening and shortening [12]. While it should be noted that breed formation is distinct from, and a secondary outcome that would follow from domestication, these changes across domestic dog breeds could represent the magnification of an initial trend.

Indeed, while such differences are often solely attributed to artificial selection, it is also likely that developmental biases are present [13] for canid skulls that direct variation toward such changes. In fact, the dog skull (both domesticated and wild), and that of other canids is known to be modular with the anterior snout region forming a separate variational module from the brain case [12,14]. Such modularity makes it possible for changes in the snout to be independent of the rest of the skull. If these patterns of modularity are deeply ancestral in canids, they could emerge as phylogenetic effects that bias the evolution of other species at a macroevolutionary scale, but also affect change at a microevolutionary scale by limiting the number of possible phenotypes. Indeed, evidence from the well-known Belyayev domestication experiments in red foxes, and which solely favoured behavioural 'tameness', was intended to mimic the selection regime during the initial domestication process of dogs but has also resulted in a number of morphological changes including a relative shortening of the snout [9,10,15].

Regarding domestic dogs, the conditions during the initiation of their domestication are largely unknown. One hypothesis suggests that behavioural changes were a driver of initial changes toward domestication. Indeed, grey wolves (Canis lupis) are social pack animals and, similar to dogs, are especially noted for their ability to convey and interpret facial expressions. Evidence suggests that wolves that were in early contact with humans developed shorter, wider skulls thought to be more interpretable by humans [5,16,17]. Alternatively, such morphological changes could simply be present due to functional demands caused by changes in diet that correspond with the presence of humans. 'Scavenging' partly processed carcasses or cooked food from humans could have reduced the stresses in wolves' jaws, thus effecting morphological change. Therefore, investigating a canid that is less social (solitary hunters but monogamous and sometimes living in small family groups) with populations showing very recent close proximity to, but with few social interactions with humans, could be particularly informative for discerning the initial primary drivers of skull shape divergence (i.e. a surrogate of the conditions that possibly initiated dog domestication).

To address these issues, we focused on testing for morphological divergence in the skull of red foxes (Vulpes vulpes) inhabiting rural and urban habitats in southern England. Starting over a century ago urban foxes have been recorded in many British cities, such as Birmingham, Bristol and London [18]. Urban foxes appear to have made a significant ecological shift as they now exploit shelter and can have upwards of 37% of their diet consisting of scavenged food [19]. In turn, urban foxes show substantially reduced home ranges in urban habitats relative to rural ones (0.4 km$^2$ versus 30 km$^2$ for urban and rural habitats, respectively), suggesting barriers to gene flow could exist and provide an opportunity to adapt to local conditions [19,20]. Indeed, previous research has suggested that urban foxes in Switzerland are somewhat genetically isolated from their rural conspecifics [21,22]. Potential morphological differences between urban and rural foxes are currently unknown but their skull provides a complex multivariate trait that could provide insights into functional differences and evolutionary mechanisms of differentiation.

Using the V. vulpes skull, we tested the general hypotheses that differential conditions (selective pressures or plasticity) between urban and rural environments would produce changes in skull morphology that reflect differences in ecology. Specifically, in line with trends found within domestication syndrome, we predicted that urban environments would favour a skull with a shorter wider snout. Additionally, in line with previous findings from mammals (but against the predictions of domestication syndrome), we predicted that urban environments would be associated with a larger brain (and hence larger brain case) due to increased cognitive demands [1]. Also, in line with other examples of habitat divergence between urban and rural environments, we hypothesized that sex differences in the divergence of skull morphology would arise between environments given differences in life history demands [23–25]. If present, we predicted that sexual dimorphism would be reduced in the urban environment in line with domestication syndrome [9]. Finally, as an alternative driver of divergence patterns we accounted for possible phylogenetic effects and developmental biases that could influence outcomes across an urban/rural habitat gradient. Specifically, we tested whether patterns of divergence between species of Vulpini were influenced by phylogeny. We also then tested whether patterns of divergence were aligned among micro- and macroevolutionary scales. Understanding these aspects of divergence in response to anthropogenic factors could greatly increase our ability to predict the responses of other animal populations to human environments, while also informing hypotheses surrounding the initiation and outcomes of domestication.

## 2. Material and methods

### (a) Selection of specimens and landmarks

A total of 111 skulls of red foxes (Vulpes vulpes) were available from London (n = 75, 38 females, 37 males) and the surrounding boroughs (n = 36, 17 males, 19 females), which are housed in the

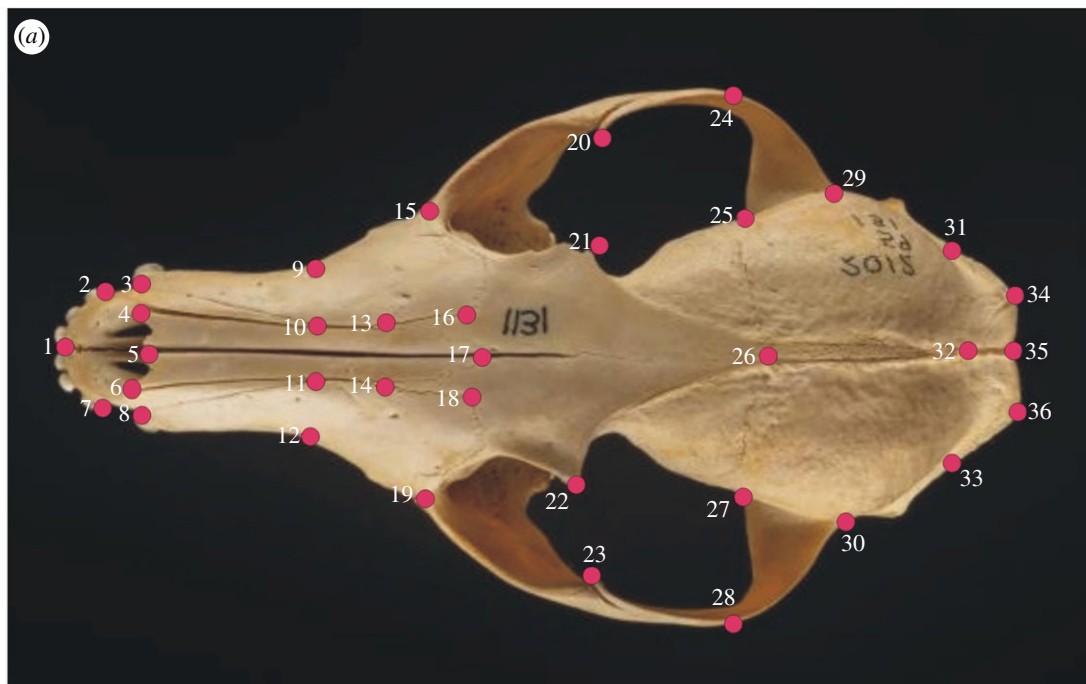

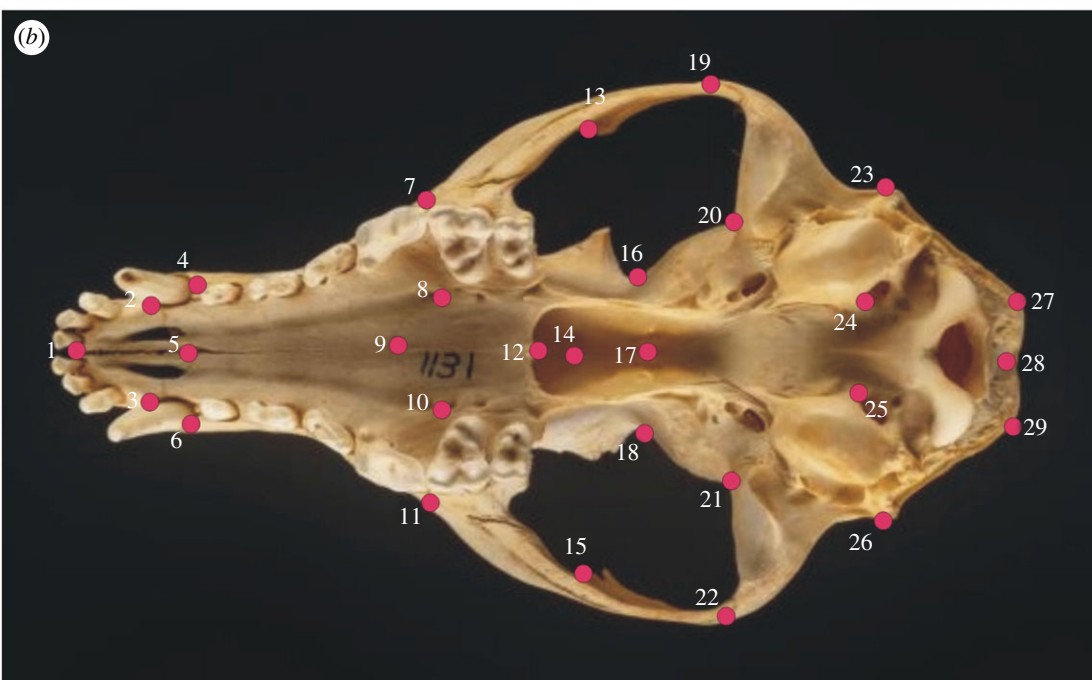

**Figure 1.** Landmarks for dorsal (*a*) and ventral (*b*) aspects of a red fox, *Vulpes vulpes*, skull.  Photographs: Neil McLean (copyright National Museums Scotland).

collections of National Museums Scotland (NMS); the location of collection, and sex of each individual are provided in electronic supplementary material, table S1. These were collected from 1971 to 1973 by Steve Harris [26]. All specimens have information about date and collecting locality, which allowed for classification of individuals into urban and rural locations although no information about relatedness was available. Locations were checked against contemporary OS maps to determine whether they were rural or urban at that time, because many locations have become urbanised since the time of collection. Urban collection sites were classified as those containing buildings, street lighting and lacking wooded areas (notably collection sites most often include a precise street name). Rural sites were dominated by wooded areas and little to no human development at time of collection. While more refined methods are available for classification of habitats [27], this was not possible with our data due to a lack of information at the time of collection. Nonetheless, our approach was in line

with previous studies assessing rural–urban differences in mammals [28]. As growth in red foxes is normally complete after a year, only adult specimens, with fused basi-sphenoid sutures, were kept for analysis to limit variation due growth allometry [29].

To quantify the morphology of fox skulls, digital photographs were taken of each specimen using a Nikon Coolpix 4500 (Nikon, Japan). Specimens were placed on modelling clay to standardize their articulation for photography. For each specimen, an image of the dorsal and ventral aspects of each skull was collected for landmarking. Briefly, the dorsal aspect of the skull was used to define 36 homologous landmarks, while the ventral aspect was used to define 29 landmarks and followed a similar protocol to Drake & Klingenberg [12] (figure 1; electronic supplementary material, table S2).

Across the Vulpini clade, we collected similar data from a further 163 specimens housed both at the National Museums Scotland and the Natural History Museum (London). This dataset

consisted of 12 species with 10 from *Vulpes* (*V. cana*, *V. chama*, *V. corsac*, *V. ferrilata*, *V. macrotis*, *V. lagopus*, *V. rueppellii*, *V. velox* and *V. zerda*), and two basal species (*Otocyon megalotis* and *Nyctereutes procyonoides*) with varying sample sizes and used the same landmarking protocol as above (electronic supplementary material, table S3). Relationships among Vulpini and the additional canids were based on a recently published supertree [30].

## (b) Statistical analysis

Landmark data for each of the ventral and dorsal aspects of each skull were corrected for variation in size and orientation using a generalized Procrustes analysis that included all specimens for each aspect. Partial warp scores, which accounted for quantitative variation in shape, were collected for each of the ventral and dorsal aspects for further statistical analysis. The steps in this analysis involved the use of tpsUtil to append all specimens into a single tps file, and tpsRelw to perform Procrustes transformation, thin-plate spline projection, and the extraction of partial warp scores [31]. Partial warp scores are amenable to multivariate statistical analyses and represent the rotation of Procrustes residuals around the Procrustes mean configuration [32].

To assess the influences of sex and habitat class on skull shape a MANOVA was performed for each skull aspect of the red fox data using base functions in R v. 3.5.0 [33]. Sex and habitat (urban/rural) and their interaction were used as explanatory variables, while partial warp scores were the shape response variables.

To determine whether the degree of sexual dimorphism differed between habitats, we compared the magnitudes of shape difference between males and females from rural and urban sites. This analysis relied on measurements of sex-based Procrustes distances defined as the square root of the sum of squared differences in the positions of the landmarks in two shapes [31]. The Procrustes distance between male and female foxes from the rural habitat was compared to the corresponding male/female Procrustes distance derived from the urban habitat using 900 bootstraps [31, p. 224]. This produced 95% confidence intervals for both urban and rural groups, with a lack of overlap indicating a significant difference. This analysis was performed using the Coordgen8 package to format landmark data files in conjunction with TwoGroup8 to perform the bootstrapping procedure [31,34].

To visualize shape variation for biological interpretation, we produced a series of deformation grids depicting the two-dimensional effects of sex and habitat on skull shape for red foxes. These deformation grids were based on canonical variables derived from our MANOVA models using the *candisc* package in R v. 3.5.0. [35]. Specifically, while canonical variables are traditionally limited to a one-way MANOVA design, this package allowed for the generalization of our two-way MANOVA designs. Therefore, our canonical scores for each of sex and habitat take account of the other factor in the model. Deformation grids for sex and habitat effects were created using these canonical scores as an independent variable in a multivariate regression on coordinate data using tpsRelw [36].

## (c) Comparative approaches

We assessed whether phylogenetic effects influenced fox skull shape. Upon image collection it was apparent that a wide degree of size variation was present among species of foxes. Therefore, to quantify and account for allometry in Vulpini skull shape, we performed a Procrustes ANOVA between centroid size and skull shape. We found a significant effect of allometry (dorsal aspect $r^2 = 0.11$, $F = 20.48$, $p = 0.001$; ventral aspect: $r^2 = 0.12$, $F = 22.11$, $p = 0.001$). Thus, we then performed a regression of shape on geometric centroid size for both dorsal and ventral aspects of the skulls to generate an allometry-minimized landmark dataset based on residuals [29]. Finally, to minimize effects from differential

sample sizes across species, we calculated mean landmark configurations for each Vulpini species, the two basal species, and then performed a principal component analysis (PCA, see electronic supplementary material, table S4) on both the individual and mean Procrustes-transformed shape data to quantify variation among skulls. We used the geomorph package (v. 3.0.1) in R to conduct tests for allometry, and to perform PCA analysis following general Procrustes superimposition [37].

All phylogenetic comparative methods were performed using a time-calibrated, species-level supertree of the Carnivora [30]. We extracted the relationships of the 12 Vulpini species from this supertree, for which we had dorsal and ventral morphometric data, and pruned all remaining taxa using the ape package in R [38].

To assess whether divergence across the urban/rural habitat axis was similar to trends found across the phylogeny we performed a series of steps. First, we used the canonical axis of habitat divergence generated above (in *candisc*) using each of the ventral and dorsal aspects (figure 1). This canonical axis and its scores represented microevolution across the urban/rural habitats while taking account of sexual dimorphism. Second, to determine the major axis of variation for Vulpini (i.e. macroevolution), we extracted the first principal component from the mean shape data of the Vulpini clade. Third, we used a multivariate version of Blomberg's K to estimate the degree of phylogenetic signal across the Vulpini clade in our PC score data (axes 1–6) from the dorsal and ventral aspects [39]. Blomberg's K measures phylogenetic signal by quantifying the amount of observed variance in dorsal and ventral PC scores relative to variance expected under Brownian motion. K ranges from 0, whereby no phylogenetic signal is detected and closely related taxa exhibit traits that, on average, are not more similar than more distantly related taxa, to infinity. When $K = 1$ the trait exhibits strong phylogenetic signal and is evolving under a model of Brownian motion. When $K > 1$ closely related taxa exhibit trait values more similar than would be expected under Brownian motion [39]. We tested whether K significantly differed from 0 (i.e. a sign of no phylogenetic signal) by comparing our value of K to a null distribution of K values generated via 1000 simulations on a star phylogeny, which serves to remove or eliminate phylogenetic signal by rescaling branch lengths [39]. We used the *K.mult* function in the R package phylocurve (v. 2.0.9) to conduct our multivariate assessment of Blomberg's K [40]. Performing evolutionary analyses on a dataset with a small number of species can result in greater error rates depending on data structure [41–43]. To determine the degree of statistical power in our analysis of K, we report the value of estimated power between the simulated data and our own given by the *K.mult* function [40].

It was qualitatively apparent that the magnitude of divergence was several fold greater among species across Vulpini than between the urban/rural habitat axis. However, both types of divergence (micro- and macroevolution) could follow a common trajectory. Therefore, we compared the major microevolutionary trajectory of skull shape divergence with the major trajectory of macroevolutionary divergence. Quantitatively, this involved extracting the main trajectories for each type of divergence (macro- and micro-). For divergence between urban and rural habitats, the canonical scores, calculated using habitat as a grouping variable (and controlling for sex variation), were used to provide a microevolutionary trajectory. To represent divergence among Vulpini, we used the major axis of divergence (i.e. PC1) derived from the landmark dataset comprised of each species' mean shape.

Specifically, because of clear differences in magnitude comparing trajectories of micro- and macroevolutionary divergence required a scale-free approach. To derive a scale-free vector of microevolutionary divergence for comparison in shape space, we regressed the Procrustes superimposed landmark data from the dorsal and ventral aspects of urban/rural populations on

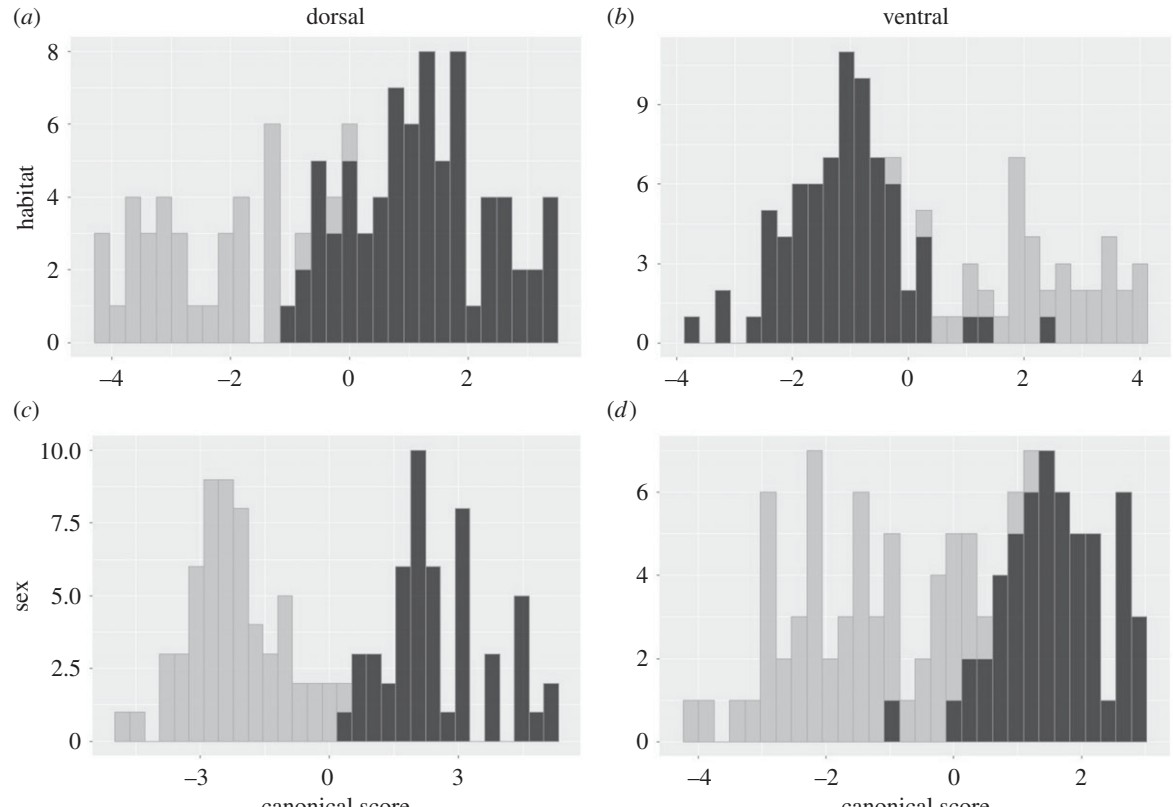

**Figure 2.** Frequency histograms depicting the statistical discrimination of habitat and sex-based differences in dorsal and ventral red fox skull shape. In (*a,b*), the habitat-based differences are depicted for the dorsal and ventral views, respectively, (light grey = rural, black = urban). In (*c,d*) the sex differences from the dorsal and ventral views are shown respectively (light grey = female, black = male).

**Table 1.** The effects of habitat, sex and their interaction on the shape of red fox (*Vulpes vulpes*) skulls from ventral and dorsal aspects as indicated by MANOVA.

| aspect | | d.f. | Pillai's trace | approx. F | Num DF | Den DF | *p*-value |
|---|---|---|---|---|---|---|---|
| ventral | habitat | 1 | 0.73 | 2.77 | 53 | 55 | <0.001 |
| factor | sex | 1 | 0.70 | 2.43 | 53 | 55 | <0.001 |
| | habitat×sex | 1 | 0.61 | 1.59 | 53 | 55 | 0.045 |
| | residuals | 107 | | | | | |
| dorsal | habitat | 1 | 0.76 | 1.86 | 67 | 40 | 0.019 |
| | sex | 1 | 0.85 | 3.48 | 67 | 40 | <0.001 |
| | habitat×sex | 1 | 0.73 | 1.64 | 67 | 40 | 0.049 |
| | residuals | 106 | | | | | |

their habitat-derived canonical axis [31, p. 257]. Similarly, the vector of macroevolutionary divergence was calculated by regressing Vulpini landmark data against PC1. The scale-free observed angle between these vectors for micro- and macroevolutionary divergence (for both dorsal and ventral aspects) was then calculated as the *arc cosine*. We then ran 900 bootstraps with replacement for each group (urban/rural and Vulpini) independently to produce 95% confidence intervals. The observed angle between micro- and macroevolutionary divergence was compared against the confidence interval of angles to determine whether it differed from random processes (i.e. did the observed angle lie outside the confidence interval?). These procedures were performed using standard routines within the software Regress8 [34]. We also performed complementary approaches through an alternative procedure based upon linear model

evaluation with a randomized residual permutation procedure [44]. While allowing for similar comparisons of trajectory this approach also allowed for tests of differences in the magnitude of evolutionary divergence along a common trajectory. Specifically, this was performed using our landmark data with habitat and vectors derived from our PC1 and DFA scores as explanatory variables and using 1000 permutations using the *pairwise* function within the RRPP package in R [44].

## 3. Results and discussion

We found strong evidence that skull shape was different between urban and rural habitats. For both the ventral and dorsal aspects, habitat had a large effect on shape (table 1

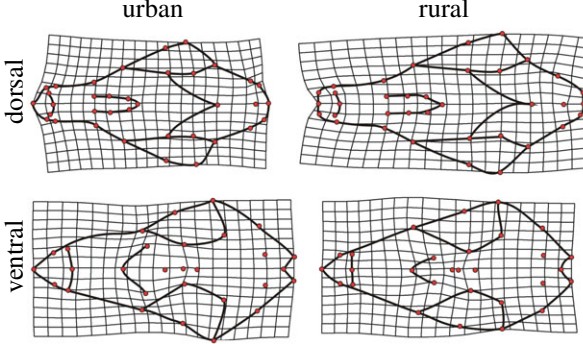

**Figure 3.** Skull shape variation in red foxes (*Vulpes vulpes*) in relation to urban and rural habitats from the dorsal and ventral aspects. Trends are magnified by 3× to enhance the interpretation of shape variation. Note the snout (LMs 1–14 in the dorsal aspect, 1–10 in the ventral view) containing the maxillary region and nasal regions, and the braincase (LMs 25–36 in the dorsal aspect, 16–29 in the ventral aspect) containing the sagittal crest.

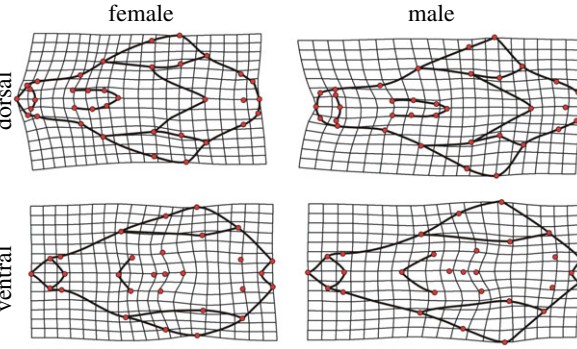

**Figure 4.** Sex-based differences in the skull shape of red foxes (*Vulpes vulpes*) from dorsal and ventral aspects. Trends in shape variation are magnified 3× to enhance the interpretation of shape variation.

and figure 2). Habitat also interacted with sex in both views, although with a strong but slightly smaller effect on shape relative to habitat. Furthermore, sex alone had a major effect on shape, especially within the dorsal aspect, where its impact was larger than habitat (table 1). The degree of sexual dimorphism also differed between habitats, with the dorsal aspect showing a significant 28% reduction in dimorphism in the urban habitat (95% CIs did not cross zero). Anatomically we found widespread differences in skull shape between habitats, with urban foxes having a noticeably shortened wider snout with a reduced maxillary region relative to rural foxes (figure 3). However, the tip of the snout, which is comprised of the premaxillary and nasal regions, showed some degree of widening in urban foxes, which was especially evident from the ventral view. Finally, the sagittal crest was extended posteriorly in urban foxes, while the zygomatic region was relatively reduced in terms of both length and width, along with the braincase. Many of these shape changes could be related to the development of jaw muscles [45,46]. While an extended sagittal crest would indicate an increased area of attachment of the temporalis muscle and indicate a higher bite force, a gracile zygomatic arch would also indicate a reduced masseter muscle in urban foxes. Indeed, finite-element modelling of biting in canids has demonstrated that the zygomatic region experiences particularly high stresses [46], and so reinforcement in this region may be adaptive.

Such anatomical variation is likely to provide a number of ecologically functional differences between urban and rural populations. Firstly, a shorter snout, as found in urban foxes, should confer a higher mechanical advantage but with reduced closing speed of the jaw [46]. This may be advantageous in an urban habitat where resources are more likely to be accessed as stationary patches of discarded human foods. Furthermore, in some cases, these foods may require a greater force to access them, explaining the expanded sagittal crest in skulls of urban foxes. Consistent with this the squamous temporalis is expanded in urban foxes as indicated from the ventral aspect (figure 3). In a rural habitat, an increased jaw-closing speed would be conferred by an increase in its length and aid in capture of motile prey, e.g. voles, mice and rabbits. While having an overall smaller snout the

increased nasal region in urban foxes (figure 3, at the tip of the snout) may also reflect their ecology, which could be more dependent on olfactory cues than other senses. Contrary to our prediction the braincase appeared to be smaller in the urban habitat. While this might suggest a smaller brain (in agreement with domestication syndrome), it could possibly reflect changes in biomechanical forces on the skull [12]. Notably, the smaller braincases found in the urban environment differ from the responses of other small mammals which show increases in braincase size [28]. Nonetheless, future work should focus on determining variation in the relative proportions of soft tissues (muscle, brains) to more precisely determine functional differences and potential adaptations among urban and rural populations. However, it would also be useful to further explore morphological variation in three-dimensions to gain further insights into masseter function. This could be indicated through how the zygomatic arches still show reductions in urban populations from a different perspective than we found.

For sex, we found strong patterns of shape divergence between males and females. A more shortened, robust skull was present in females, whereby the zygomatic region was greatly reduced relative to males, which possessed a larger, more protruding, squamous temporalis and thus larger distances between the zygomatic arch and frontal bone. Notably, males displayed more elongated snouts, with reduced crania (figure 4). In relation to the patterns seen between urban and rural habitats, this suggests that females are better adapted to the potential demands of an urban environment. Indeed, selection may be stronger on females as during parental care periods female red foxes visited dens more frequently and for longer periods of time than males. This suggests that they engage with local foraging conditions more intensively relative to males, especially given the greater caloric demands placed on them during parental care [47]. This may also confer greater cognitive demands in females explaining their relatively enlarged crania. In contrast, male red foxes engage in vigilant behaviours more frequently during periods of parental care and this may involve defensive actions that favour the faster more elongate jaws we observed. If selection is driving a stronger evolutionary response to urban environments in females it could lead to an overall 'feminization' of urban populations through sexual conflict. If so, this would also be consistent with expected changes under domestication and deserves further attention.

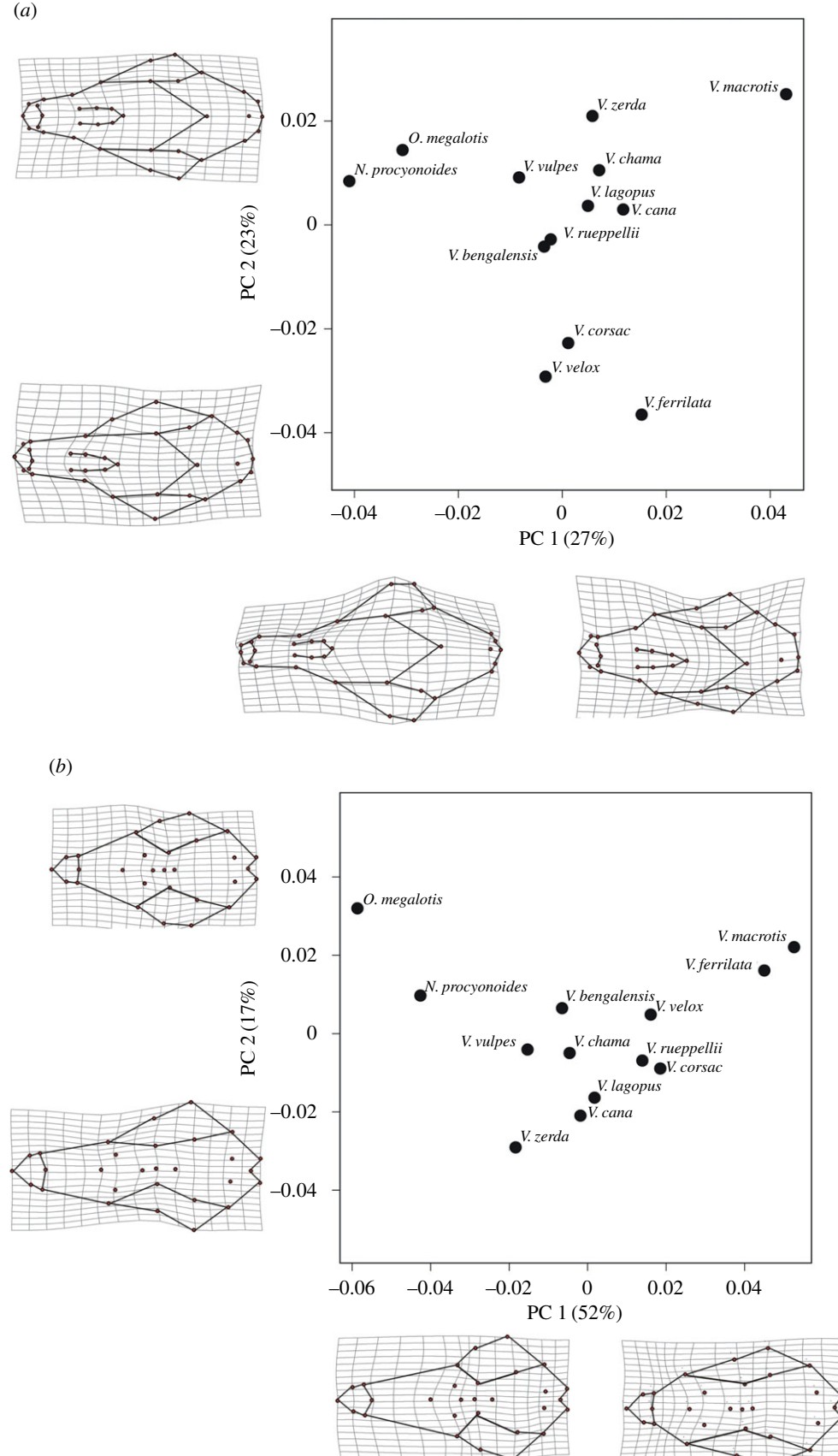

**Figure 5.** Scatterplots depicting the morphospace of the genus *Vulpes* and basal species (*O. megalotis, N. procyonoides*). The first two principal components from the dorsal (*a*) and ventral (*b*) aspects of fox skulls are portrayed with the associated shape changes for the extremes of each axis also being depicted as deformation grids.

## (a) Phenotypic trajectories of micro- versus macroevolution

For both dorsal and ventral aspects across the Vulpini, PC1 characterized morphological change that involved lengthening and shortening of the snout that was, respectively,

concomitant with a lateral widening or narrowing of the skull. Widening was especially noticeable in the dorsal view around the zygomatic arch. On PC2, it appeared that rostral width, sagittal crest length and the length of secondary palate changed together. Within Vulpini it appeared that red foxes occupied a central region of morphospace in close

proximity to basal species (figure 5). We also found low levels of phylogenetic signal in both our ventral and dorsal view PC score data. In both aspects we found no evidence to suggest $K$ significantly differed from our null hypothesis of 0 (dorsal aspect: $K.mult = 0.461$, $p = 0.356$, power $= 0.98$; ventral aspect: $K.mult = 0.545$, $p = 0.129$, power $= 0.93$), indicating both aspects lack phylogenetic signal. We note our values of $K$ were robust to the effects of low sample size as our analysis exhibited high power to detect differences among models. Interpreting which evolutionary processes may have led to Vulpini exhibiting low phylogenetic signal is difficult, as the relationship between $K$ and a number of these evolutionary processes, such as the rate of morphological evolution, genetic drift, or gene flow, is often complex [40]. Nonetheless, this finding in support of no phylogenetic signal allowed us to readily compare trajectories of macro- and microevolutionary divergence (pending evidence of heritable variation) directly from unmodified landmark data.

We found that the pattern of divergence between urban and rural habitats did not differ from the major axis of variation found in Vulpini. Specifically, observed vector values did not exceed bootstrapped confidence intervals produced from both the dorsal and ventral aspects (90° and 97° for observed angles for dorsal and ventral aspects respectively, CIs = 177° to 48°, and 108° to 32° for dorsal and ventral aspects, respectively). These results were affirmed by our additional RRPP approach, which also indicated that the magnitude of morphological change on shared trajectories was greater across the clade than between urban/rural habitats (both $p > 0.001$). Therefore, coupled with a lack of phylogenetic signal, our data suggest that the directions of evolution available to red foxes diverging between urban and rural habitats are not constrained by evolutionary history, yet they follow the same pattern as their clade. Specifically, in much the same way as divergence in red foxes between urban and rural habitats, Vulpini is mainly characterized by lengthening and shortening of the snout (figure 5). Thus, the conditions presented by recent anthropogenic habitats may favour phenotypes that play a role in the speciation of Vulpini. However, this could also indicate that developmental biases common to Vulpini are playing a role in determining phenotypic variation for contemporary evolution [9,13,48]. Future approaches implementing higher resolution 3D morphometrics could help to clarify this evidence by providing more comprehensive information about shape variation.

## (b) Wider implications and relationship to domestication

Notably, some of the craniofacial features that differ between urban and rural habitats are also similar to the effects of 'domestication syndrome' [5,6]. These can include traits such as docile behaviour, craniofacial morphology, ear floppiness, reductions in brain size, reduced sexual dimorphism and changes in pigmentation [9,48]. Specifically, in red foxes experimental domestication via selection on behavioural traits, more precisely 'friendliness' towards humans, has resulted in reduced muzzle and jaw sizes that accompany docility [15]. While not domesticated, urban foxes show reductions in muzzle size, reduced sexual dimorphism, and a narrowed braincase, and it is plausible that taking up residence in the presence of humans would favour individuals with reduced levels of fear and stress (i.e. urban tameness) as it

has in other animals [49]. Mechanistically in experimental domestication this could be traced to a reduced size and function of the adrenal glands, but how could this relate to changes in the craniofacial apparatus? Recently, Wilkins et al. [9,48] proposed a link among 'domestication' traits that can be traced to neural crest cells (NCCs). NCCs are a vertebrate-specific class of stem cells that first appear during early embryogenesis at the dorsal edge (crest) of the neural tube. These cells migrate throughout the body toward the cranium and trunk and provide the cellular precursors of many cell and tissue types, including many of the bony elements of the skull, and the adrenal medulla [50]. In line with the idea that changes in NCCs underlie these traits, a number of neural-crest-related genes have been implicated in domestication processes [48,51].

Our findings of craniofacial divergence along an urban/rural habitat axis in red foxes suggests that some phenotypic traits related to domestication are involved, and perhaps influenced by developmental bias present within Vulpini that generally funnels variation toward a long/short jaw axis [12,13,52]. While these differences may be adaptive, they could also arise from founder effects, or other random processes. Additionally, the urban environment may actually relax selection, if it provides greater food resources and a reduced need to hunt. The inclusion of additional urban/rural gradients in future studies would be useful for discerning these possibilities. Regardless, it is notable that the trajectory of divergence taken by red foxes in response to urbanization is similar to that found over the past 15 Myr of fox evolution, suggesting that biases could be having long-term effects. Phenotypically, an interesting next step would be to assess the heritability of the morphological traits we characterize here along with a suite of additional traits related to domestication. For example, and following that these patterns mirror changes during domestication, it could be possible to experimentally compare responses to behavioural stress between fox populations in these habitats and determine whether this corresponds with variation in the size of the adrenal gland. However, genomic analyses could also enable a wide array of traits to be implicated as an evolutionary factor and identified for further study. The recently assembled red fox genome has already been used to identify regions associated with tame and aggressive behaviours [51]. Genes related to neural crest activity are well characterized [50,53] and would provide an interesting inroad into the mechanisms underlying adaptive divergence in anthropogenic environments, initiators of domestication, as well as long-term macroevolutionary change.

Data accessibility. Ventral and dorsal landmark data for *Vulpes vulpes*. Ventral and dorsal landmark data for genus and ancestors. Data available from the Dryad Digital Repository: https://doi.org/10.5061/dryad.bnzs7h47c [54].

Authors' contributions. K.J.P. conceived the ideas, collected data, analysed the data and led the writing of the manuscript. A.R. and H.Z. collected data and assisted with writing and analysis. A.J.C. edited drafts and performed analyses while A.K. provided access to samples, information about them, and helped by editing drafts. S.H. donated samples to the NMS collection. K.J.P., A.R., H.Z. and A.K. contributed critically to the drafts and gave final approval for publication.

Competing interests. We declare we have no competing interests.

Funding. During the time of writing K.J.P. was supported by a grant from NERC (NE/N016734/1).

Acknowledgements. We thank Zena Timmons and Roberto Miguez for assistance with museum collections in Edinburgh and London

respectively. We thank Neil McLean for photos of the red fox skull in figure 1. We also thank W. J. Cooper for providing photos of kit foxes. The input from three anonymous reviewers greatly improved this manuscript.

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
