## [Reviewer comments · Proceedings of the Royal Society B: Biological Sciences]

Review History

RSPB-2019-1138.R0 (Original submission)

Review form: Reviewer 1

Recommendation

Accept with minor revision (please list in comments)

Scientific importance: Is the manuscript an original and important contribution to its field?

Good

General interest: Is the paper of sufficient general interest?

Good

Quality of the paper: Is the overall quality of the paper suitable?

Good

Is the length of the paper justified?

Yes

Should the paper be seen by a specialist statistical reviewer?

Yes

Do you have any concerns about statistical analyses in this paper? If so, please specify them explicitly in your report.

No

It is a condition of publication that authors make their supporting data, code and materials available - either as supplementary material or hosted in an external repository. Please rate, if applicable, the supporting data on the following criteria.

Is it accessible?

Yes

Is it clear?

Yes

Is it adequate?

Yes

Do you have any ethical concerns with this paper?

No

Comments to the Author

This is a very interesting manuscript looking at the morphological differences between urban and rural foxes that shows some intriguing parallels between urbanisation and domestication/macroevolutionary trends. This is a really valuable dataset and I am surprised it hasn't been studied more since its collection in the 1970s. On the whole, I think the manuscript is presented and written well. The context given in the introduction seems appropriate and the methods, as far as I can tell, have been carried out correctly. For the most part, the results are presented well and the interpretation of those results is justified. I just have a few comments:

Line 16: Add 'be' after 'found to'

Line 35: 'cities should' feels like odd phrasing to me. Maybe 'cities may'?

Line 112: It would be good to see a table of specimens, with sex and habitat (and location/grid reference) in the supplementary info.

Line 114: What were the criteria for classifying specimens as urban or rural?

Line 123: Why did you choose to analyse both dorsal and ventral views? I'm not sure that they provide different information from one another. Why was a lateral view not included? This could have provided useful information on cranial and rostral height. Indeed, why not use 3D data - was it simply a lack of access to 3D digitising equipment? In your discussion, I think you need to at least acknowledge the limitations of 2D data and how it might have affected your results.

Line 218: I appreciate that the MANOVAs on partial warp scores have indicated significant differences between habitats and sexes, but I would have liked to see the distribution of the data across a morphospace to get an idea of how clearly split the groups are.

Line 227: Clarify the way in which the sagittal crest has been extended - anteriorly or posteriorly?

Line 230: change 'could' to 'would'

Line 230: 'a greater need for muscle attachment' is a slightly odd construction. 'could indicate an increased area of attachment of the temporalis' is what you mean I think.

Line 238: Harder foods do not need a 'higher mechanical advantage to access' - they simply require greater force.

Line 238: An 'expanded sagittal crest' would not necessarily increase mechanical advantage of the temporalis.

Line 239: What do you mean by an expanded posterior region of the zygomatic arch? Are you referring to a greater width of the zygomatic process of the squamosal? And how does this fit with your statement earlier that urban foxes have a reduced zygomatic region (line 228)?

Line 241: Actually Santana and Dumont are somewhat cautious about the supposedly high stresses in the zygomatic region in their bat FE models, suggesting they might be modelling artifacts. A better reference might be Wroe et al (2007; Proc B 274: 2819-2828) who actually look at the masticatory biomechanics of a canid (the dingo).

Line 250: In what way are the zygomatic region and the cranium reduced in females and males respectively?

Line 262: I was pleased to see the morphospaces in Figure 4, but I think you need to describe the results of the PCA – how are the species distributed, what is the shape change along the axes, where do red foxes plot compared to other *Vulpes* species?

Line 262: Why was only PC1 tested for phylogenetic signal? Blomberg's K can be calculated for multivariate data and the *Vulpes* specimens seem to spread across PC2 mostly.

Line 271: Add 'of' after 'pattern'

Supplemental table 1: Why are the species in this order? It is neither alphabetical nor phylogenetic as far as I can see. Also, *Vulpes rueppellii* needs to be italicised.

Supplemental table 2: 'palantine' should be 'palatine' in ventral landmarks 8, 10 and 12.

Review form: Reviewer 2

Recommendation

Reject – article is scientifically unsound

Scientific importance: Is the manuscript an original and important contribution to its field?

Acceptable

General interest: Is the paper of sufficient general interest?

Good

Quality of the paper: Is the overall quality of the paper suitable?

Marginal

Is the length of the paper justified?

No

Should the paper be seen by a specialist statistical reviewer?

Yes

Do you have any concerns about statistical analyses in this paper? If so, please specify them explicitly in your report.

Yes

It is a condition of publication that authors make their supporting data, code and materials available - either as supplementary material or hosted in an external repository. Please rate, if applicable, the supporting data on the following criteria.

Is it accessible?

N/A

Is it clear?

N/A

Is it adequate?

Yes

Do you have any ethical concerns with this paper?

No

Comments to the Author

Line 49, refs 5-6: This statement and the references are correct, but it should be noted here that recent work has questioned the universality of the syndrome, and this is important as it should lead to qualifications and caveats when making generalisations. Still, the patterns in CANIDS, to which foxes belong, is there, so this is not fatal to the ideas presented here, but it is as stated a misrepresentation of the current knowledge not to refer to this work.

Sánchez-Villagra MR, Geiger M, Schneider RA. 2016. Taming the neural crest: A developmental perspective on the origins of morphological covariation in domesticated mammals. *Royal Society Open Science* 3 160107.

Concerning the statements in lines 60-61 on modularity favouring change in snout independent of the rest and so on, this statement should be qualified in view of both Drake and Klingenberg 2010 and Curth et al 2017 (ref below), which show that the degree of modularity in dogs is no different from that of wolves or other carnivorans, and no association with disparity was found. It is still possible and reasonable to state that modularity facilitate what is stated there, but this is not something peculiar to dogs – this should be made clear, because it weakens the argument of dev bias of line 58 as presented.

Curth, Stefan & S. Fischer, Martin & Kupczik, Kornelius. (2017). Patterns of integration in the canine skull: An inside view into the relationship of the skull modules of domestic dogs and wolves. *Zoology*. 125. 10.1016/j.zool.2017.06.002.

Line 70 at the end: 'changes' – do you mean morphological changes? If so, please state that. From what follows it seems you mean exclusively skull changes. I would be specific there then. In the paragraph starting line 69, I would add the neural crest hypothesis as a third alternative, and present the three hypothesis as not exclusive from each other.

Line 78: I understand the logic of using foxes to discern between the two hypothesis – but it is a little of a stretch, and if we consider the third, alternative hypothesis (neural crest), also considered by the authors, the logic here is not so strong. This should be acknowledged.

Line 81: 'morphological divergence' – this refers exclusively to skull shape – it would be fair to be explicit about that from the start. This is perfectly fine, but it should be made clear.

Line 97: braincase, one word

Line 98: for the 'increased cognitive demands' leading to 'larger braincase' reference 1 is provided, which is on butterflies – please change that. Actually, citing the works of Kruska and other here would not fully fix this idea, as it is contested this relation when it comes to subtle differences and here it would be more relevant to differentiate among different parts of the brain.

Lines 125-126: unclear what is meant here with 'basal species'

Line 128: species of what?

Lines 171-176 and what follows:

I understand how having a baseline of comparison using a phylogenetic bracket is useful – so the changes in the foxes can be put in context by comparing with several Carnivora, 'correcting' or considering phylogeny. But what we have here are samples of foxes in cities versus rural habitats from 1972-1973 and then samples of several species of Carnivora - so the statement in lines 175-176 'To assess whether divergence across the urban/rural habitat axis was influenced by phylogenetic effects...' makes not much sense to me. Maybe it is the way it is expressed. How could phylogeny influence such as differentiation? Surely phylogeny influences everything and we need to account for it, but here some fancy analysis does not fit I would say, at least as it is

presented.

I do not see then the significance of the lines 268-270 to start with.

I recommend to add relevant commas in sentences in the following lines: 224, 251, 257, 259.

I evaluate the discussion in page 10, lines 235-247 as original, as these functional aspects have not been discussed in the recent literature dealing with changes in skull shape driven by domestication (eg Evin and colleagues work on pigs, Sánchez-Villagra et al. work on domestic skull growth).

Lines 284-285: the developmental bias hypothesis here is not clear.

Line 297: Wilkins, not Wilkens. See other recommended literature here. Lines 303-304: there is also a recent review by Wilkins AS 2017 REVISITING TWO HYPOTHESES ON THE DOMESTICATION SYNDROME" IN LIGHT OF GENOMIC DATA (Russian Journal, in EN) which is quite relevant here.

This is a recent and relevant paper discussing the parallels with island and domestication, neural crest hypothesis, etc., treated in this manuscript.

van der Geer AAE. 2019. Effect of isolation on coat colour polymorphism of Polynesian rats in Island Southeast Asia and the Pacific. PeerJ 7:e6894 <https://doi.org/10.7717/peerj.6894>

Review form: Reviewer 3

Recommendation

Major revision is needed (please make suggestions in comments)

Scientific importance: Is the manuscript an original and important contribution to its field?

Acceptable

General interest: Is the paper of sufficient general interest?

Acceptable

Quality of the paper: Is the overall quality of the paper suitable?

Acceptable

Is the length of the paper justified?

Yes

Should the paper be seen by a specialist statistical reviewer?

No

Do you have any concerns about statistical analyses in this paper? If so, please specify them explicitly in your report.

Yes

It is a condition of publication that authors make their supporting data, code and materials available - either as supplementary material or hosted in an external repository. Please rate, if applicable, the supporting data on the following criteria.

Is it accessible?

N/A

Is it clear?

N/A

Is it adequate?

N/A

Do you have any ethical concerns with this paper?

No

Comments to the Author

Main problems

1. Absence of evidence is not evidence of absence. Would their value of K, if the parameter equalled the estimate, put an important phylogenetical spoke into the wheels of the later analysis? (p11 lines 263-265 suggest rather little power). Specifically,

- (a) the authors more than once interpret absence of evidence as evidence of absence. Apart from justifying their later analyses, the passage in lines 307-309 is a clear example.
- (b) they rely on no phylogenetic signal for a whole set of analyses
- (c) it seems likely they have little power in estimating K

I note that quite a lot of the paper seems to depend on these analyses and doubtful interpretations.

2. Misinterpretation of confidence of intervals

p11 line 275. The main cause of width of confidence is inadequacy of data (not enough, not informative enough about the parameter). To interpret confidence intervals as though they reflect the world, rather than the imperfections of our perceptions of it, is to try to make a silk purse out of a sow's ear. (In fact the statement is ambiguous about the comparison being made between the confidence intervals. Is it (i) in position, so one CI is somewhat to the right of another CI (ii) in position, but one CI and the other don't overlap (this would usually be recordable in a significance test), or (iii) in width, so one is wider than the other, without regard to position?)

Minor thoughts

1. The authors don't say whether the Habit*Sex interaction is of the kind that indicates a different relationship e.g. multiplicativity, which might in principle be abolished by a transformation; or a qualitative difference in how the sexes respond to habitats.

2. Family group problems are possible, causing non-independence of the samples, depending on how the skulls were collected. I haven't looked at any of the referenced papers, but there is the possibility that there are sampling biases of which it would be helpful to be aware.

2a Pursuing that thought, the urban/rural divide will in fact be a number of different urban places and a number of different rural places. Most obviously, the urban will divide by conurbation. In pinciple, we could regard these conurbations as random effects. Alternatively, a more granular analysis could show that the same pattern occurs in the different conurbations (perhaps by absence of interactions with conurbation in an ANOVA or MANOVA). However, most "accidental sampling" studies will have potential problems of this kind, and I don't want to over-stress them.

3. In the MANOVA tables on p19, p-values are given with a > instead a <, Thus, for example, ">0.001" should read "<0.001".

4. The sample sizes of the species (p25) are very uneven, making the species-level analyses not so persuasive (as the balance of within-species and between-species causes of variation in the mean will be different for the different species, making homoscedasticity assumptions less credible).

5. Would it not be natural to control for species in the MANOVAs on p19?

6. In the MANOVA on p19, both interactions are given p-values of 0.05. Looking them up myself from F, DF1 and DF2, they are both just less than 0.05. I see the value in not giving too many decimal places for approximate values (as the F is approximate), but on the other hand, the reader should somehow be informed that the exact lookup of the approximate result is actually less than 0.05 rather than, say, 0.054. It is one source of p-value-inflation that need not be worried about.

Decision letter (RSPB-2019-1138.R0)

05-Aug-2019

Dear Dr Parsons:

I am writing to inform you that your manuscript RSPB-2019-1138 entitled "Skull morphology diverges between urban and rural populations of red foxes and mirrors patterns of domestication and macroevolution" has, in its current form, been rejected for publication in Proceedings B.

This action has been taken on the advice of referees, who have recommended that substantial revisions are necessary. With this in mind we would be happy to consider a resubmission, provided the comments of the referees are fully addressed. However please note that this is not a provisional acceptance. Indeed, the reviews are very split and it is not certain that the same referees would accept re-review.

In your revision process, please take a second look at how open your science is; our policy is that all data involved with the study should be made openly accessible-- see: <https://royalsociety.org/journals/ethics-policies/data-sharing-mining/>
Insufficient sharing of data can delay or even cause rejection of a paper.

Sincerely,
Professor John Hutchinson, Editor
<mailto:proceedingsb@royalsociety.org>

Associate Editor

Board Member: 1

Comments to Author:

Thank you for the opportunity to review this paper. The impacts of humanisation of the environment on macro-level anatomy and functional morphology is a highly topical subject, and I found this analysis of skull shape in foxes (using a very nice anatomical data set) to be very interesting. Firstly, I would like to apologise for the extremely lengthy delay with peer review. As the authors will see, reviewers 1 and 2 provided very different views of the manuscript and this necessitated a third review, which unfortunately took a long time to find. Based on my own reading of the manuscript, and the views of the reviewers, it would be my recommendation that the paper be rejected in its current form, but that the authors have the opportunity to revise and resubmit to Proceedings B. In particular I think the majority of points raised by reviewers 1 and 3 in their comments require careful consideration.

Reviewer(s)' Comments to Author:

Referee: 1

Comments to the Author(s)

This is a very interesting manuscript looking at the morphological differences between urban and rural foxes that shows some intriguing parallels between urbanisation and domestication/ macroevolutionary trends. This is a really valuable dataset and I am surprised it hasn't been studied more since its collection in the 1970s. On the whole, I think the manuscript is presented and written well. The context given in the introduction seems appropriate and the methods, as far as I can tell, have been carried out correctly. For the most part, the results are presented well and the interpretation of those results is justified. I just have a few comments:

Line 16: Add 'be' after 'found to'

Line 35: 'cities should' feels like odd phrasing to me. Maybe 'cities may'?

Line 112: It would be good to see a table of specimens, with sex and habitat (and location/grid reference) in the supplementary info.

Line 114: What were the criteria for classifying specimens as urban or rural?

Line 123: Why did you choose to analyse both dorsal and ventral views? I'm not sure that they provide different information from one another. Why was a lateral view not included? This could have provided useful information on cranial and rostral height. Indeed, why not use 3D data – was it simply a lack of access to 3D digitising equipment? In your discussion, I think you need to at least acknowledge the limitations of 2D data and how it might have affected your results.

Line 218: I appreciate that the MANOVAs on partial warp scores have indicated significant differences between habitats and sexes, but I would have liked to see the distribution of the data across a morphospace to get an idea of how clearly split the groups are.

Line 227: Clarify the way in which the sagittal crest has been extended – anteriorly or posteriorly?

Line 230: change 'could' to 'would'

Line 230: 'a greater need for muscle attachment' is a slightly odd construction. 'could indicate an increased area of attachment of the temporalis' is what you mean I think.

Line 238: Harder foods do not need a 'higher mechanical advantage to access' – they simply require greater force.

Line 238: An 'expanded sagittal crest' would not necessarily increase mechanical advantage of the temporalis.

Line 239: What do you mean by an expanded posterior region of the zygomatic arch? Are you referring to a greater width of the zygomatic process of the squamosal? And how does this fit with your statement earlier that urban foxes have a reduced zygomatic region (line 228)?

Line 241: Actually Santana and Dumont are somewhat cautious about the supposedly high stresses in the zygomatic region in their bat FE models, suggesting they might be modelling

artifacts. A better reference might be Wroe et al (2007; Proc B 274: 2819-2828) who actually look at the masticatory biomechanics of a canid (the dingo).

Line 250: In what way are the zygomatic region and the cranium reduced in females and males respectively?

Line 262: I was pleased to see the morphospaces in Figure 4, but I think you need to describe the results of the PCA – how are the species distributed, what is the shape change along the axes, where do red foxes plot compared to other *Vulpes* species?

Line 262: Why was only PC1 tested for phylogenetic signal? Blomberg's K can be calculated for multivariate data and the *Vulpes* specimens seem to spread across PC2 mostly.

Line 271: Add 'of' after 'pattern'

Supplemental table 1: Why are the species in this order? It is neither alphabetical nor phylogenetic as far as I can see. Also, *Vulpes rueppellii* needs to be italicised.

Supplemental table 2: 'palantine' should be 'palatine' in ventral landmarks 8, 10 and 12.

Referee: 2

Comments to the Author(s)

Line 49, refs 5-6: This statement and the references are correct, but it should be noted here that recent work has questioned the universality of the syndrome, and this is important as it should lead to qualifications and caveats when making generalisations. Still, the patterns in CANIDS, to which foxes belong, is there, so this is not fatal to the ideas presented here, but it is as stated a misrepresentation of the current knowledge not to refer to this work.

Sánchez-Villagra MR, Geiger M, Schneider RA. 2016. Taming the neural crest: A developmental perspective on the origins of morphological covariation in domesticated mammals. *Royal Society Open Science* 3 160107.

Concerning the statements in lines 60-61 on modularity favouring change in snout independent of the rest and so on, this statement should be qualified in view of both Drake and Klingenberg 2010 and Curth et al 2017 (ref below), which show that the degree of modularity in dogs is no different from that of wolves or other carnivorans, and no association with disparity was found. It is still possible and reasonable to state that modularity facilitate what is stated there, but this is not something peculiar to dogs – this should be made clear, because it weakens the argument of dev bias of line 58 as presented.

Curth, Stefan & S. Fischer, Martin & Kupczik, Kornelius. (2017). Patterns of integration in the canine skull: An inside view into the relationship of the skull modules of domestic dogs and wolves. *Zoology*. 125. 10.1016/j.zool.2017.06.002.

Line 70 at the end: 'changes' – do you mean morphological changes? If so, please state that. From what follows it seems you mean exclusively skull changes. I would be specific there then.

In the paragraph starting line 69, I would add the neural crest hypothesis as a third alternative, and present the three hypothesis as not exclusive from each other.

Line 78: I understand the logic of using foxes to discern between the two hypothesis – but it is a little of a stretch, and if we consider the third, alternative hypothesis (neural crest), also considered by the authors, the logic here is not so strong. This should be acknowledged.

Line 81: 'morphological divergence' – this refers exclusively to skull shape – it would be fair to be explicit about that from the start. This is perfectly fine, but it should be made clear.

Line 97: braincase, one word

Line 98: for the 'increased cognitive demands' leading to 'larger braincase' reference 1 is provided, which is on butterflies – please change that. Actually, citing the works of Kruska and other here would not fully fix this idea, as it is contested this relation when it comes to subtle differences and here it would be more relevant to differentiate among different parts of the brain.

Lines 125-126: unclear what is meant here with 'basal species'
 Line 128: species of what?

Lines 171-176 and what follows:

I understand how having a baseline of comparison using a phylogenetic bracket is useful – so the changes in the foxes can be put in context by comparing with several Carnivora, 'correcting' or considering phylogeny. But what we have here are samples of foxes in cities versus rural habitats from 1972-1973 and then samples of several species of Carnivora - so the statement in lines 175-176 'To assess whether divergence across the urban/rural habitat axis was influenced by phylogenetic effects...' makes not much sense to me. Maybe it is the way it is expressed. How could phylogeny influence such as differentiation? Surely phylogeny influences everything and we need to account for it, but here some fancy analysis does not fit I would say, at least as it is presented.

I do not see then the significance of the lines 268-270 to start with.

I recommend to add relevant commas in sentences in the following lines: 224, 251, 257, 259.

I evaluate the discussion in page 10, lines 235-247 as original, as these functional aspects have not been discussed in the recent literature dealing with changes in skull shape driven by domestication (eg Evin and colleagues work on pigs, Sánchez-Villagra et al. work on domestics skull growth).

Lines 284-285: the developmental bias hypothesis here is not clear.

Line 297: Wilkins, not Wilkens. See other recommended literature here. Lines 303-304: there is also a recent review by Wilkins AS 2017 REVISITING TWO HYPOTHESES ON THE DOMESTICATION SYNDROME" IN LIGHT OF GENOMIC DATA (Russian Journal, in EN) which is quite relevant here.

This is a recent and relevant paper discussing the parallels with island and domestication, neural crest hypothesis, etc., treated in this manuscript.

van der Geer AAE. 2019. Effect of isolation on coat colour polymorphism of Polynesian rats in Island Southeast Asia and the Pacific. PeerJ 7:e6894 <https://doi.org/10.7717/peerj.6894>

Referee: 3

Comments to the Author(s)

Main problems

1. Absence of evidence is not evidence of absence. Would their value of K, if the parameter equalled the estimate, put an important phylogenetical spoke into the wheels of the later analysis? (p11 lines 263-265 suggest rather little power). Specifically,
 - (a) the authors more than once interpret absence of evidence as evidence of absence. Apart from justifying their later analyses, the passage in lines 307-309 is a clear example.
 - (b) they rely on no phylogenetic signal for a whole set of analyses
 - (c) it seems likely they have little power in estimating K

I note that quite a lot of the paper seems to depend on these analyses and doubtful interpretations.

2. Misinterpretation of confidence of intervals

p11 line 275. The main cause of width of confidence is inadequacy of data (not enough, not informative enough about the parameter). To interpret confidence intervals as though they reflect the world, rather than the imperfections of our perceptions of it, is to try to make a silk purse out

of a sow's ear. (In fact the statement is ambiguous about the comparison being made between the confidence intervals. Is it (i) in position, so one CI is somewhat to the right of another CI (ii) in position, but one CI and the other don't overlap (this would usually be recordable in a significance test), or (iii) in width, so one is wider than the other, without regard to position?)

Minor thoughts

1. The authors don't say whether the Habit*Sex interaction is of the kind that indicates a different relationship e.g. multiplicativity, which might in principle be abolished by a transformation; or a qualitative difference in how the sexes respond to habitats.

2. Family group problems are possible, causing non-independence of the samples, depending on how the skulls were collected. I haven't looked at any of the referenced papers, but there is the possibility that there are sampling biases of which it would be helpful to be aware.

2a Pursuing that thought, the urban/rural divide will in fact be a number of different urban places and a number of different rural places. Most obviously, the urban will divide by conurbation. In principle, we could regard these conurbations as random effects. Alternatively, a more granular analysis could show that the same pattern occurs in the different conurbations (perhaps by absence of interactions with conurbation in an ANOVA or MANOVA). However, most "accidental sampling" studies will have potential problems of this kind, and I don't want to over-stress them.

3. In the MANOVA tables on p19, p-values are given with a > instead a <, Thus, for example, ">0.001" should read "<0.001".

4. The sample sizes of the species (p25) are very uneven, making the species-level analyses not so persuasive (as the balance of within-species and between-species causes of variation in the mean will be different for the different species, making homoscedasticity assumptions less credible).

5. Would it not be natural to control for species in the MANOVAs on p19?

6. In the MANOVA on p19, both interactions are given p-values of 0.05. Looking them up myself from F, DF1 and DF2, they are both just less than 0.05. I see the value in not giving too many decimal places for approximate values (as the F is approximate), but on the other hand, the reader should somehow be informed that the exact lookup of the approximate result is actually less than 0.05 rather than, say, 0.054. It is one source of p-value-inflation that need not be worried about.

Author's Response to Decision Letter for (RSPB-2019-1138.R0)

See Appendix A.

RSPB-2020-0142.R0

Review form: Reviewer 1 (Philip Graham Cox)

Recommendation

Accept as is

Scientific importance: Is the manuscript an original and important contribution to its field?
Excellent

General interest: Is the paper of sufficient general interest?
Excellent

Quality of the paper: Is the overall quality of the paper suitable?
Excellent

Is the length of the paper justified?
Yes

Should the paper be seen by a specialist statistical reviewer?
No

Do you have any concerns about statistical analyses in this paper? If so, please specify them explicitly in your report.
No

It is a condition of publication that authors make their supporting data, code and materials available - either as supplementary material or hosted in an external repository. Please rate, if applicable, the supporting data on the following criteria.

Is it accessible?
Yes

Is it clear?
Yes

Is it adequate?
Yes

Do you have any ethical concerns with this paper?
No

Comments to the Author

I am satisfied that the authors have addressed all my comments appropriately. As far as I can tell they have also addressed the comments of the other reviewers in a sensible fashion.

Review form: Reviewer 3

Recommendation
Reject - article is scientifically unsound

Scientific importance: Is the manuscript an original and important contribution to its field?
Acceptable

General interest: Is the paper of sufficient general interest?
Acceptable

Quality of the paper: Is the overall quality of the paper suitable?
Poor

Is the length of the paper justified?

Yes

Should the paper be seen by a specialist statistical reviewer?

No

Do you have any concerns about statistical analyses in this paper? If so, please specify them explicitly in your report.

Yes

It is a condition of publication that authors make their supporting data, code and materials available - either as supplementary material or hosted in an external repository. Please rate, if applicable, the supporting data on the following criteria.

Is it accessible?

N/A

Is it clear?

N/A

Is it adequate?

N/A

Do you have any ethical concerns with this paper?

No

Comments to the Author

I'm afraid that the statistics is still gobbledeygook to me. This may be my unfamiliarity with the methods, but the editor has to ask how widely the paper should be comprehensible. The idea of showing that the urban-rural effect is in line with the generic line seems very sensible, and I have no problem with that. But what is said in the MS about the machinery employed makes no sense to me.

First, I return to the question of confidence intervals. The authors disagree with me in their response to referees. Specifically, they write "We disagree that we have misinterpreted our analysis based on confidence intervals. The width of a confidence interval can indeed be due to inadequacy of data as the reviewer asserts, but it can also be due to real variation in the data." To return to basics, an X% confidence interval is defined for a parameter of a statistical model as the range of values that cannot be rejected at the X% level. My difficulty with the use of confidence intervals in the MS include (i) the angle is not a parameter of the model, but something measured after the fit (ii) a confidence interval becomes narrower and narrower (approaching zero width) as the amount of data increases. It is (as its definition states) an assertion about our uncertainty about a parameter. I do not see how a comparison between the width of two confidence intervals in two different datasets can reflect on anything other than our relative uncertainty about the parameters on the basis of the information in one model versus the other (iii) the confidence intervals themselves seem not to be given, as I read lines 307--308. I can only understand this is saying that (for example) the width of the confidence interval for dorsal-rural/urban is 178 degrees, while the width for dorsal-genus-variation confidence interval is 48 degrees. The confidence interval itself should be a range from one angle to another, and it seems odd not to give that range. It may well be that the term confidence interval is used in this area of biology in a nonstandard way -- perhaps others have done so too. But I am perfectly clear that the MS's use of "confidence interval" is incompatible with the standard definition.

Second, my questions about estimation of K have stimulated a new power analysis, mentioned around line 205 and provided in lines 293-296. The MS states that their analysis exhibits "high power to detect differences among models", and gives significant p-values. But it doesn't say

which models were being compared, and so the statement provides little information. Generally, a power analysis would have to specify the magnitude of the deviation from the null hypothesis that was being tested. That is, one might say a test has only 30% power against a specified small deviation, but 90% power against some specified larger deviation. The MS quotes power levels, but without stating the parameter values (presumably for K) used to provide them. The "critical test statistic" mentioned on lines 295-296 is pointless to give, as it is not at all explained. Thus, the power analyses need, at least, more explanation.

Third, there is the question of supporting null hypotheses. The authors now claim to provide support for a null hypothesis on line 309. They also affirm a null hypothesis ("indicating both views lack a phylogenetic signal") on line 292. The logic of hypothesis testing simply doesn't provide for supporting or affirming a null hypothesis. The type of statement presumably desired would be statistically justifiably provided by a confidence interval for some parameter, showing that the CI is narrow around some particular value. Of course, that relies on a judgment of narrowness, which the author will make, but which the reader will be able to judge for himself or herself. Whether a mean ± 0.1 is close enough to the particular value not to matter, rather depends on the context, and on the question at hand. But there is no sense in using the term "null hypothesis" for the particular value.

Finally, I didn't follow fully all the stuff about regressing landmarks on canonical variables, but I think I get the gist. (I presume that canonical variables were produced from the landmark data.) My question here is: in order to show that the direction of change in urban/rural is the same as within the genus, why is one not comparing the canonical variables? In particular, the regression seems to be asking how the urban/rural data differ from *their* canonical axis, and how the inter-specific data differ from *their* canonical axis. Why are those deviations within each dataset relevant to whether the canonical variables are the same? I can see that the deviations within a dataset could be used to estimate the certainty of the canonical variable loadings, and that this could in principle lead to a test of whether the two canonical variables differ. BUT, comparing the angles of the intra-dataset comparisons doesn't take into account the difference between the canonical variables, which surely must be relevant.

The authors may feel that they have important anatomical points to make about fox skulls and domestication, and that the statistics is really unimportant. It may well be that a much simpler statistical argument would be fully acceptable to justify their biological conclusions. I have no reason to doubt their biological conclusions that the urban/rural changes are in line with the inter-specific changes. But I hope I've explained why the statistical aspect of the paper has a number of problematic elements.

Some minor points:

214. Drop the comma? (Or indeed move earlier in sentence)

219. Is their approach scale free? PCA isn't!

312. Should it be $p < 0.001$?? rather than $>$;

316 "way to divergence" : the "to" reads very oddly -- as?

Reference 38 is a non-functioning URL. I was hoping to find some explanation of their calculations with regress8.

Review form: Reviewer 4

Recommendation

Major revision is needed (please make suggestions in comments)

Scientific importance: Is the manuscript an original and important contribution to its field?

Excellent

General interest: Is the paper of sufficient general interest?

Excellent

Quality of the paper: Is the overall quality of the paper suitable?

Good

Is the length of the paper justified?

Yes

Should the paper be seen by a specialist statistical reviewer?

No

Do you have any concerns about statistical analyses in this paper? If so, please specify them explicitly in your report.

No

It is a condition of publication that authors make their supporting data, code and materials available - either as supplementary material or hosted in an external repository. Please rate, if applicable, the supporting data on the following criteria.

Is it accessible?

No

Is it clear?

N/A

Is it adequate?

N/A

Do you have any ethical concerns with this paper?

No

Comments to the Author

This manuscript is about cranial shape variation in urban vs. rural red foxes. The authors found that there are significant differences in skull shape between urban and rural foxes and that these shape changes are consistent with the ones found in an interspecific comparison among other *Vulpini*. This is an interesting study on the subject of urban adaptation, which is still far to less extensively investigated, despite its rising importance in a steadily urbanizing world.

The data set is exceptionally extensive and, as far as I can judge, the applied methods are sound. However, I feel that the manuscript lacks clarity in many instances (see my detailed comments below). The terminology and the descriptions related to taxonomy, systematics, and concepts of evolution, development, domestication, and breed formation are oftentimes quite peculiar and unusual compared to subject-specific literature, as well as too simplified, in some parts at the brink to non-correct (again, see my detailed comments below). Further, I would strongly recommend to include further references in various instances (also as indicated below) and to

extend the discussion. For example, the authors discuss only adaptive concepts, without taking into account non-adaptive processes, which might explain their findings.

Detailed suggestions for edits:

L. 15: Genus name *Vulpes* in upper case.

L. 20-21: I think your sampling of the genus *Vulpes* is actually not complete, as stated here. According to the standard reference work by Wilson & Reeder (Mammal species of the world, 2005, 3rd edition), there is also *Vulpes pallida*, which you did not sample. Please rephrase, or specify later which reference for *Vulpes* you were using.

L. 23 & L. 40: Do you mean divergence “of skull shape” between urban and rural habitats? Please make clear what you mean. Otherwise it sounds as if the habitats themselves would diverge.

L. 24-25: This sentence basically repeats the previous one. Maybe delete?

L. 25-26: Why “additionally”? Dogs are one example of many where this shortening of the snout has occurred during domestication. Plus, please be aware that domestication is not equal to breed formation, so the term “breed” should not be used here.

L. 37-40: Maybe add here, that selection could not only be different across a suite of functionally salient traits, but also across clades.

L. 48-49: It would be appropriate to cite also the original paper by the people who came up with the term ‘domestication syndrome’: Wilkins et al. 2014 (I think it’s your reference 44).

L. 48 & 51: What do you mean by “stereotypical”? That similar changes occur in different domestic species? Please clarify.

L. 50-53: This list is not exhaustive and I would recommend to include “for example” or something similar.

Paragraph on domestication syndrome: I would strongly recommend to also cite the new paper by Lord et al. 2019, who critically discuss the ‘domestication syndrome’: Lord, K. A., Larson, G., Coppinger, R. P., & Karlsson, E. K. (2019). The history of farm foxes undermines the animal domestication syndrome. *Trends in Ecology & Evolution*.

Further, and I have already mentioned this above: the selection for a shorter snout in some dog breeds (as described e.g. by Drake & Klingenberg 2010) has nothing to do with the domestication syndrome, but with breed formation. This means, with selection for specific traits subsequent to domestication. Domestication and breed formation are two very different processes and should not be mixed.

L. 55: There is no evidence that the extensive morphological change in dogs has been particularly fast. See e.g. Geiger, M., & Sánchez-Villagra, M. R. (2018). Similar rates of morphological evolution in domesticated and wild pigs and dogs. *Frontiers in zoology*, 15(1), 23.

I would further strongly recommend to refer to the domestic form of dogs as ‘domestic dogs’ or the like, not just ‘dogs’. This could be confused with you referring to the clade Canidae, which is usually also referred to as ‘dogs’.

L.58: Developmental biases ‘within’ dog skulls is a phrasing I’ve never heard before. I would delete this.

Additionally: I think artificial selection for these traits and developmental biases are not mutually exclusive. It may be that head shape development is biased in one way or the other, leading to a

limited number of possible morphotypes within an evolutionary lineage. However, such morphotypes, e.g. as seen in bulldog or greyhound, would not be prevalent in these breeds if it wasn't for artificial selection, i.e., people who were selecting for these traits as they occurred.

L. 61: Developmental modularity does not 'favour' changes in the snout independent of the rest of the skull, but it 'makes possible' (or something like this). These are developmental patterns, not "conscious" processes.

L. 65-66: '...shows a similar trend in skull shape.' What exactly? Skull shape change? Please specify.

L. 67: What do you mean by "mimic ancestral condition for dogs?" It is not clear which condition you mean. I would suggest to put write something in the line of "mimic the selection regime during the initial domestication process of dogs".

You could also cite here this study on mice, in which selection for tameness was more similar to the scenario in dogs and other commensal domesticates (and has led to a shortening of the head): Geiger, M., Sánchez-Villagra, M. R., & Lindholm, A. K. (2018). A longitudinal study of phenotypic changes in early domestication of house mice. *Royal Society open science*, 5(3), 172099.

L. 69: "during" instead of "for"?

L. 70: "driver of initial changes" of what? Please specify.

L. 78: Why do you point out that foxes are non-social? Does this have an implication for this study? In fact, I would not regard foxes as non-social per-se, as they are sometimes living in family groups and are showing complex social behaviour. I agree that the behaviour may be less complex than in wolves (e.g., solitary vs. pack hunting), but still not non-social.

L. 84: City foxes are not only known from Great Britain, but also from Central Europe: Gloor, S., Bontadina, F., Hegglin, D., & Deplazes 2001. The rise of urban fox populations in Switzerland. *Mammalian Biology*, 66, 155-164.

L. 89: In this study on urban and rural foxes in Switzerland, it has been shown that urban foxes are indeed somewhat isolated from their rural conspecifics: Wandeler, P., Funk, S. M., Largiadere, C. R., Gloor, S., & Breitenmoser, U. (2003). The city-fox phenomenon: Genetic consequences of a recent colonization of urban habitat. *Molecular Ecology*, 12(3), 647-656.

L.94ff: You predict that rostral shape in urban foxes would follow the line of the domestication syndrome, but for brain size and sexual dimorphism it would be the opposite. This is contradictory. After all, in urban house finches, it was found that brain and eye size is smaller compared to rural conspecifics (Hutton & McGraw 2016). Further, in general, domestication is associated with a decrease of sexual dimorphism.

Hutton, P., & McGraw, K. J. (2016). Urban-rural differences in eye, bill, and skull allometry in house finches (*Haemorrhous mexicanus*). *Integrative and comparative biology*, 56(6), 1215-1224.

L. 100-102: The comparison rural vs. urban is intraspecific. I do not see why these should be biased by phylogenetic relatedness and why this analysis is important here.

L. 108 ff. (Materials and Methods): Would it be possible to provide the number of specimens in each habitat (rural and urban) as well as the sex in a table or written out in the text in the main manuscript, please? This would be crucial for the interpretation of the results, especially to see if there were e.g. equal numbers of males and females for each habitat. It's good to have the supplementary table with all the specimens, but a summary would also be important for the main text.

L. 111: I would recommend to write: Which are “housed” in the collections of the...

L. 113: All specimens “have”

L. 124-125: So I guess the fusion of the basisphenoid synchondrosis is associate with age? Could you please provide a reference here?

L. 129: Maybe “aspects” instead of “views”?

L. 129-131: This is a peculiar phrasing. Dorsal and ventral aspects do not “provide” landmarks. Rather, you have defined these landmarks on these aspects. Please rephrase.

Figure 1:

- Delete “the” and “genus” in “the *Vulpes* genus”. It’s clear that *Vulpes* is a genus.

- Landmark descriptions are actually in supplementary table 3.

Supplementary Table 3.

- Landmark 5: delete duplication of words.

- What is the difference between landmarks 2 and 3 (and 7 and 8)? Please describe 2 and 7 more accurately.

L. 135: The last part of this sentence seems not to belong there. Rephrase?

L. 136: This is actually Supp. Table 2, not 1. I think this table, or at least the names of the species, should go in the main text.

Further, what do you mean by “additional canids”? As far as I see, you have sampled almost the whole clade *Vulpes* (for the one exception see my comment above) as well as the basal *Otocyon* and *Nyctereutes*. This means you have basically sampled the clade *Vulpini*. Please make this clear, as this phylogenetic context is important to understand the implications of your analysis.

L. 162: what do you mean by “they”? Do you mean you want to test if similar shape changes as seen in interspecific comparison are also prevalent in an intraspecific (urban vs. rural) comparison?

L. 165 ff.: which one is the dependent and which the independent variable? Please specify. Please write again what you mean by “shape”. Partial warp scores?

Also, I do not see why you performed an ANOVA first. Aren’t both of your variables continuous? You were performing a regression analysis anyway, so this tells you the same thing. Or do I miss something here?

Please rephrase: there is no such a thing as “effect of allometry on shape”. It would be “effect of size on shape”, which is allometry or allometric scaling.

L. 170: You say you did this for *Vulpes* species, however, in the Figure showing the results of PCA, there are also the other two non-*Vulpes* species of *Vulpini*. Please rephrase.

L. 171: this is actually supplementary table 4, not 3.

L. 181: It is not 12 *Vulpes* species (see above), but 12 species of *Vulpini*. Please change (also in remainder of the text).

L. 197: Remove brackets around (BM). I would suggest to not abbreviate this term anyway. Every abbreviation makes a text harder to read and this specific term only occurs three times in the manuscript.

L.161-161 and L.184-185: These two paragraphs have a similar introductory sentence. This is confusing and I would suggest to condensate.

L. 209: You did not perform a 'phylogenetic analysis'. This would be the analysis of phylogenetic relationships between clades. Please rephrase.

L. 208-211: This is confusing. You are performing the test to determine if phylogeny has an effect, but saying here that there is no need to account for a phylogenetic signal. Maybe I do not get the point here, but I think it would be helpful to rephrase.

L. 224: delete one "run"

L. 237: It's a bit of a stretch to write that skull shape is influenced by rural and urban habitat. I would recommend writing something along the line of: "skull shape is different in foxes urban vs. rural foxes".

Figure 2: Would it be possible to choose the colours so that they can also be discerned in black and white? Also, the numbers on the axes are very small and I would recommend to make them bigger.

Figure 3 and 4: I would recommend to NOT magnify the shape differences. This is misleading. Especially because you write on L. 242 that you found "extensive" differences. I doubt that these differences are that extensive if they are not magnified (but I would be happy to be proven wrong).

L. 249 & 250: Do you mean temporalis and masseter "muscle"? Please complement.

L. 261: Do you mean increased jaw "closing" speed? Please specify.

L. 263,264: Isn't the olfactory sense also important in rural areas? Please explain why you think it is more important in urban areas.

L. 270 ff.: You found that in an interspecific comparison, body size differences influence skull shape in Vulpini. Could the shape differences seen in male and female red foxes also be related to size? If yes, I would be careful to argue with adaptive arguments here. Could you elaborate on this please?

L. 284: Are these the results corrected for influence of differences in body size? Or the raw data? Please specify, as this is important for the interpretation (especially because PC1 in non-size corrected data is usually strongly correlated with body size)

And again, it is not just the genus *Vulpes*, but the clade Vulpini.

Figure 5: *O. megalotis* and *N. procyonides* are not ancestors of the *Vulpes* species! This is as incorrect as if one would say: Chips are ancestors of humans. They are "basal" or "sister taxa". The ancestor of *Vulpes* is probably extinct. Please rephrase.

Please report the percentage of variation for each PC on the axis.

L. 288-289: I disagree. On PC1, *V. vulpes* is actually the species which is the most (or second most) negative.

Further, you do not discuss PC2. How do you interpret these results?

L. 330-332: "Urban tameness" is indeed a termed - as you surely know - and I would recommend to cite e.g. the following article or a related one:

Uchida, K., Suzuki, K. K., Shimamoto, T., Yanagawa, H., & Koizumi, I. (2019). Decreased vigilance or habituation to humans? Mechanisms on increased boldness in urban animals. *Behavioral Ecology*, 30(6), 1583-1590.

L. 342: You are discussing adaptation to an urban environment. But what about non-adaptive processes? For example, the urban population might be characterised by a small founder population (the few individuals which are able to cope with the urban setting), which is again associated with genetic drift. I think that the possibility of the observed patterns being the result of random processes should be discussed. Further, I think it would be worthwhile mentioning the effects of "ecological release" on skull shape: urban foxes do not have to hunt as frequently as their rural conspecifics, which reduces selection pressures, which in turn may increase variability of skull shape. These are all non-adaptive effects which should be discussed.

Decision letter (RSPB-2020-0142.R0)

25-Feb-2020

I am writing to inform you that this version of your manuscript RSPB-2020-0142 entitled "Skull morphology diverges between urban and rural populations of red foxes mirroring patterns of domestication and macroevolution" has, in its current form, been rejected for publication in *Proceedings B*.

This action has been taken on the advice of referees, who have recommended that substantial revisions are necessary. With this in mind we would be happy to consider a resubmission, provided the comments of the referees are fully addressed. However please note that this is not a provisional acceptance.

Please find below the comments made by the referees, not including confidential reports to the Editor, which I hope you will find useful.

Please note that this decision may (or may not) have taken into account confidential comments.

In your revision process, please take a second look at how open your science is; our policy is that

ALL (maximally inclusive) data involved with the study should be made openly accessible, fully enabling re-use, replication and transparency-- see:
<https://royalsociety.org/journals/ethics-policies/data-sharing-mining/>
Insufficient sharing of data can delay or even cause rejection of a paper.

Sincerely,
Professor John Hutchinson, Editor
mailto: proceedingsb@royalsociety.org

Associate Editor Board Member: 1
Comments to Author:

As mentioned previously, I think this paper deals with topical subject, and the authors have accumulated a very nice anatomical data set with which to address the questions surrounding urbanisation on animal form-function. The reviewers generally share this positive outlook on both the topic and the attempt to analyse the data. However, reviewer 2 suggests some reasonable improvements to improve clarity and smooth out the rough edges on certain interpretations. Reviewer 3 remains concerned about some of the statistical analyses and particularly the way they are rationalised and presented. It is my recommendation that the authors be given the opportunity to resubmit a modified version to incorporate the changes suggested by Reviewers 2 & 3.

Reviewer(s)' Comments to Author:

Referee: 1

Comments to the Author(s).

I am satisfied that the authors have addressed all my comments appropriately. As far as I can tell they have also addressed the comments of the other reviewers in a sensible fashion.

Referee: 4

Comments to the Author(s).

This manuscript is about cranial shape variation in urban vs. rural red foxes. The authors found that there are significant differences in skull shape between urban and rural foxes and that these shape changes are consistent with the ones found in an interspecific comparison among other *Vulpini*. This is an interesting study on the subject of urban adaptation, which is still far to less extensively investigated, despite its rising importance in a steadily urbanizing world.

The data set is exceptionally extensive and, as far as I can judge, the applied methods are sound. However, I feel that the manuscript lacks clarity in many instances (see my detailed comments below). The terminology and the descriptions related to taxonomy, systematics, and concepts of evolution, development, domestication, and breed formation are oftentimes quite peculiar and unusual compared to subject-specific literature, as well as too simplified, in some parts at the brink to non-correct (again, see my detailed comments below). Further, I would strongly recommend to include further references in various instances (also as indicated below) and to extend the discussion. For example, the authors discuss only adaptive concepts, without taking into account non-adaptive processes, which might explain their findings.

Detailed suggestions for edits:

L. 15: Genus name *Vulpes* in upper case.

L. 20-21: I think your sampling of the genus *Vulpes* is actually not complete, as stated here. According to the standard reference work by Wilson & Reeder (Mammal species of the world,

2005, 3rd edition), there is also *Vulpes pallida*, which you did not sample. Please rephrase, or specify later which reference for *Vulpes* you were using.

L. 23 & L. 40: Do you mean divergence “of skull shape” between urban and rural habitats”? Please make clear what you mean. Otherwise it sounds as if the habitats themselves would diverge.

L. 24-25: This sentence basically repeats the previous one. Maybe delete?

L. 25-26: Why “additionally”? Dogs are one example of many where this shortening of the snout has occurred during domestication. Plus, please be aware that domestication is not equal to breed formation, so the term “breed” should not be used here.

L. 37-40: Maybe add here, that selection could not only be different across a suite of functionally salient traits, but also across clades.

L. 48-49: It would be appropriate to cite also the original paper by the people who came up with the term ‘domestication syndrome’: Wilkins et al. 2014 (I think it’s your reference 44).

L. 48 & 51: What do you mean by “stereotypical”? That similar changes occur in different domestic species? Please clarify.

L. 50-53: This list is not exhaustive and I would recommend to include “for example” or something similar.

Paragraph on domestication syndrome: I would strongly recommend to also cite the new paper by Lord et al. 2019, who critically discuss the ‘domestication syndrome’: Lord, K. A., Larson, G., Coppinger, R. P., & Karlsson, E. K. (2019). The history of farm foxes undermines the animal domestication syndrome. *Trends in Ecology & Evolution*.

Further, and I have already mentioned this above: the selection for a shorter snout in some dog breeds (as described e.g. by Drake & Klingenberg 2010) has nothing to do with the domestication syndrome, but with breed formation. This means, with selection for specific traits subsequent to domestication. Domestication and breed formation are two very different processes and should not be mixed.

L. 55: There is no evidence that the extensive morphological change in dogs has been particularly fast. See e.g. Geiger, M., & Sánchez-Villagra, M. R. (2018). Similar rates of morphological evolution in domesticated and wild pigs and dogs. *Frontiers in zoology*, 15(1), 23.

I would further strongly recommend to refer to the domestic form of dogs as ‘domestic dogs’ or the like, not just ‘dogs’. This could be confused with you referring to the clade Canidae, which is usually also referred to as ‘dogs’.

L.58: Developmental biases ‘within’ dog skulls is a phrasing I’ve never heard before. I would delete this.

Additionally: I think artificial selection for these traits and developmental biases are not mutually exclusive. It may be that head shape development is biased in one way or the other, leading to a limited number of possible morphotypes within an evolutionary lineage. However, such morphotypes, e.g. as seen in bulldog or greyhound, would not be prevalent in these breeds if it wasn’t for artificial selection, i.e., people who were selecting for these traits as they occurred.

L. 61: Developmental modularity does not ‘favour’ changes in the snout independent of the rest of the skull, but it ‘makes possible’ (or something like this). These are developmental patterns, not “conscious” processes.

L. 65-66: ‘...shows a similar trend in skull shape.’ What exactly? Skull shape change? Please specify.

L. 67: What do you mean by “mimic ancestral condition for dogs?” It is not clear which condition you mean. I would suggest to put write something in the line of “mimic the selection regime during the initial domestication process of dogs”.

You could also cite here this study on mice, in which selection for tameness was more similar to the scenario in dogs and other commensal domesticates (and has led to a shortening of the head): Geiger, M., Sánchez-Villagra, M. R., & Lindholm, A. K. (2018). A longitudinal study of phenotypic changes in early domestication of house mice. *Royal Society open science*, 5(3), 172099.

L. 69: “during” instead of “for”?

L. 70: “driver of initial changes” of what? Please specify.

L. 78: Why do you point out that foxes are non-social? Does this have an implication for this study? In fact, I would not regard foxes as non-social per-se, as they are sometimes living in family groups and are showing complex social behaviour. I agree that the behaviour may be less complex than in wolves (e.g., solitary vs. pack hunting), but still not non-social.

L. 84: City foxes are not only known from Great Britain, but also from Central Europe: Gloor, S., Bontadina, F., Hegglin, D., & Deplazes 2001. The rise of urban fox populations in Switzerland. *Mammalian Biology*, 66, 155-164.

L. 89: In this study on urban and rural foxes in Switzerland, it has been shown that urban foxes are indeed somewhat isolated from their rural conspecifics: Wandeler, P., Funk, S. M., Lurgiader, C. R., Gloor, S., & Breitenmoser, U. (2003). The city-fox phenomenon: Genetic consequences of a recent colonization of urban habitat. *Molecular Ecology*, 12(3), 647-656.

L.94ff: You predict that rostral shape in urban foxes would follow the line of the domestication syndrome, but for brain size and sexual dimorphism it would be the opposite. This is contradictory. After all, in urban house finches, it was found that brain and eye size is smaller compared to rural conspecifics (Hutton & McGraw 2016). Further, in general, domestication is associated with a decrease of sexual dimorphism.

Hutton, P., & McGraw, K. J. (2016). Urban-rural differences in eye, bill, and skull allometry in house finches (*Haemorrhous mexicanus*). *Integrative and comparative biology*, 56(6), 1215-1224.

L. 100-102: The comparison rural vs. urban is intraspecific. I do not see why these should be biased by phylogenetic relatedness and why this analysis is important here.

L. 108 ff. (Materials and Methods): Would it be possible to provide the number of specimens in each habitat (rural and urban) as well as the sex in a table or written out in the text in the main manuscript, please? This would be crucial for the interpretation of the results, especially to see if there were e.g. equal numbers of males and females for each habitat. It’s good to have the supplementary table with all the specimens, but a summary would also be important for the main text.

L. 111: I would recommend to write: Which are “housed” in the collections of the...

L. 113: All specimens “have”

L. 124-125: So I guess the fusion of the basisphenoid synchondrosis is associate with age? Could you please provide a reference here?

L. 129: Maybe “aspects” instead of “views”?

L. 129-131: This is a peculiar phrasing. Dorsal and ventral aspects do not “provide” landmarks. Rather, you have defined these landmarks on these aspects. Please rephrase.

Figure 1:

- Delete “the” and “genus” in “the *Vulpes* genus”. It’s clear that *Vulpes* is a genus.
- Landmark descriptions are actually in supplementary table 3.

Supplementary Table 3.

- Landmark 5: delete duplication of words.
- What is the difference between landmarks 2 and 3 (and 7 and 8)? Please describe 2 and 7 more accurately.

L. 135: The last part of this sentence seems not to belong there. Rephrase?

L. 136: This is actually Supp. Table 2, not 1. I think this table, or at least the names of the species, should go in the main text.

Further, what do you mean by “additional canids”? As far as I see, you have sampled almost the whole clade *Vulpes* (for the one exception see my comment above) as well as the basal *Otocyon* and *Nyctereutes*. This means you have basically sampled the clade *Vulpini*. Please make this clear, as this phylogenetic context is important to understand the implications of your analysis.

L. 162: what do you mean by “they”? Do you mean you want to test if similar shape changes as seen in interspecific comparison are also prevalent in an intraspecific (urban vs. rural) comparison?

L. 165 ff.: which one is the dependent and which the independent variable? Please specify. Please write again what you mean by “shape”. Partial warp scores?

Also, I do not see why you performed an ANOVA first. Aren’t both of your variables continuous? You were performing a regression analysis anyway, so this tells you the same thing. Or do I miss something here?

Please rephrase: there is no such a thing as “effect of allometry on shape”. It would be “effect of size on shape”, which is allometry or allometric scaling.

L. 170: You say you did this for *Vulpes* species, however, in the Figure showing the results of PCA, there are also the other two non-*Vulpes* species of *Vulpini*. Please rephrase.

L. 171: this is actually supplementary table 4, not 3.

L. 181: It is not 12 *Vulpes* species (see above), but 12 species of *Vulpini*. Please change (also in remainder of the text).

L. 197: Remove brackets around (BM). I would suggest to not abbreviate this term anyway. Every abbreviation makes a text harder to read and this specific term only occurs three times in the manuscript.

L.161-161 and L.184-185: These two paragraphs have a similar introductory sentence. This is confusing and I would suggest to condensate.

L. 209: You did not perform a ‘phylogenetic analysis’. This would be the analysis of phylogenetic relationships between clades. Please rephrase.

L. 208-211: This is confusing. You are performing the test to determine if phylogeny has an effect, but saying here that there is no need to account for a phylogenetic signal. Maybe I do not get the point here, but I think it would be helpful to rephrase.

L. 224: delete one "run"

L. 237: It's a bit of a stretch to write that skull shape is influenced by rural and urban habitat. I would recommend writing something along the line of: "skull shape is different in foxes urban vs. rural foxes".

Figure 2: Would it be possible to choose the colours so that they can also be discerned in black and white? Also, the numbers on the axes are very small and I would recommend to make them bigger.

Figure 3 and 4: I would recommend to NOT magnify the shape differences. This is misleading. Especially because you write on L. 242 that you found "extensive" differences. I doubt that these differences are that extensive if they are not magnified (but I would be happy to be proven wrong).

L. 249 & 250: Do you mean temporalis and masseter "muscle"? Please complement.

L. 261: Do you mean increased jaw "closing" speed? Please specify.

L. 263.264: Isn't the olfactory sense also important in rural areas? Please explain why you think it is more important in urban areas.

L. 270 ff.: You found that in an interspecific comparison, body size differences influence skull shape in Vulpini. Could the shape differences seen in male and female red foxes also be related to size? If yes, I would be careful to argue with adaptive arguments here. Could you elaborate on this please?

L. 284: Are these the results corrected for influence of differences in body size? Or the raw data? Please specify, as this is important for the interpretation (especially because PC1 in non-size corrected data is usually strongly correlated with body size)

And again, it is not just the genus *Vulpes*, but the clade Vulpini.

Figure 5: *O. megalotis* and *N. procyonides* are not ancestors of the *Vulpes* species! This is as incorrect as if one would say: Chips are ancestors of humans. They are "basal" or "sister taxa". The ancestor of *Vulpes* is probably extinct. Please rephrase.

Please report the percentage of variation for each PC on the axis.

L. 288-289: I disagree. On PC1, *V. vulpes* is actually the species which is the most (or second most) negative.

Further, you do not discuss PC2. How do you interpret these results?

L. 330-332: "Urban tameness" is indeed a termed – as you surely know – and I would recommend to cite e.g. the following article or a related one:

Uchida, K., Suzuki, K. K., Shimamoto, T., Yanagawa, H., & Koizumi, I. (2019). Decreased vigilance or habituation to humans? Mechanisms on increased boldness in urban animals. *Behavioral Ecology*, 30(6), 1583-1590.

L. 342: You are discussing adaptation to an urban environment. But what about non-adaptive processes? For example, the urban population might be characterised by a small founder population (the few individuals which are able to cope with the urban setting), which is again

associated with genetic drift. I think that the possibility of the observed patterns being the result of random processes should be discussed. Further, I think it would be worthwhile mentioning the effects of "ecological release" on skull shape: urban foxes do not have to hunt as frequently as their rural conspecifics, which reduces selection pressures, which in turn may increase variability of skull shape. These are all non-adaptive effects which should be discussed.

Referee: 3

Comments to the Author(s).

I'm afraid that the statistics is still gobbledeygook to me. This may be my unfamiliarity with the methods, but the editor has to ask how widely the paper should be comprehensible. The idea of showing that the urban-rural effect is in line with the generic line seems very sensible, and I have no problem with that. But what is said in the MS about the machinery employed makes no sense to me.

First, I return to the question of confidence intervals. The authors disagree with me in their response to referees. Specifically, they write "We disagree that we have misinterpreted our analysis based on confidence intervals. The width of a confidence interval can indeed be due to inadequacy of data as the reviewer asserts, but it can also be due to real variation in the data." To return to basics, an X% confidence interval is defined for a parameter of a statistical model as the range of values that cannot be rejected at the X% level. My difficulty with the use of confidence intervals in the MS include (i) the angle is not a parameter of the model, but something measured after the fit (ii) a confidence interval becomes narrower and narrower (approaching zero width) as the amount of data increases. It is (as its definition states) an assertion about our uncertainty about a parameter. I do not see how a comparison between the width of two confidence intervals in two different datasets can reflect on anything other than our relative uncertainty about the parameters on the basis of the information in one model versus the other (iii) the confidence intervals themselves seem not to be given, as I read lines 307--308. I can only understand this is saying that (for example) the width of the confidence interval for dorsal-rural/urban is 178 degrees, while the width for dorsal-genus-variation confidence interval is 48 degrees. The confidence interval itself should be a range from one angle to another, and it seems odd not to give that range. It may well be that the term confidence interval is used in this area of biology in a nonstandard way -- perhaps others have done so too. But I am perfectly clear that the MS's use of "confidence interval" is incompatible with the standard definition.

Second, my questions about estimation of K have stimulated a new power analysis, mentioned around line 205 and provided in lines 293-296. The MS states that their analysis exhibits "high power to detect differences among models", and gives significant p-values. But it doesn't say which models were being compared, and so the statement provides little information. Generally, a power analysis would have to specify the magnitude of the deviation from the null hypothesis that was being tested. That is, one might say a test has only 30% power against a specified small deviation, but 90% power against some specified larger deviation. The MS quotes power levels, but without stating the parameter values (presumably for K) used to provide them. The "critical test statistic" mentioned on lines 295-296 is pointless to give, as it is not at all explained. Thus, the power analyses need, at least, more explanation.

Third, there is the question of supporting null hypotheses. The authors now claim to provide support for a null hypothesis on line 309. They also affirm a null hypothesis ("indicating both views lack a phylogenetic signal") on line 292. The logic of hypothesis testing simply doesn't provide for supporting or affirming a null hypothesis. The type of statement presumably desired would be statistically justifiably provided by a confidence interval for some parameter, showing that the CI is narrow around some particular value. Of course, that relies on a judgment of narrowness, which the author will make, but which the reader will be able to judge for himself or herself. Whether a mean ± 0.1 is close enough to the particular value not to matter, rather

depends on the context, and on the question at hand. But there is no sense in using the term "null hypothesis" for the particular value.

Finally, I didn't follow fully all the stuff about regressing landmarks on canonical variables, but I think I get the gist. (I presume that canonical variables were produced from the landmark data.) My question here is: in order to show that the direction of change in urban/rural is the same as within the genus, why is one not comparing the canonical variables? In particular, the regression seems to be asking how the urba/rural data differ from *their* canonical axis, and the how the inter-specific data differ from *their* canonical axis. Why are those deviations within each dataset relevant to whether the canonical variables are the same? I can see that the deviations within a dataset could be used to estimate the certainty of the canonical variable loadings, and that this could in principle lead to a test of whether the two canonical variables differ. BUT, comparing the angles of the intra-dataset comparisons doesn't take into account the different between the canonical variables, which surely must be relevant.

The authors may feel that they have important anatomical points to make about fox skulls and domestication, and that the statistics is really unimportant. It may well be that a much simpler statistical argument would be fully acceptable to justify their biological conclusions. I have no reason to doubt their biological conclusions that the urban/rural changes are in line with the inter-specific changes. But I hope I've explained why the statistical aspect of the paper has a number of problematic elements.

Some minor points:

214. Drop the comma? (Or indeed move earlier in sentence)

219. Is their approach scale free? PCA isn't!

312. Should it be $p < 0.001$?? rather than $>$

316 "way to divergence" : the "to" reads very oddly -- as?

Reference 38 is a non-functioning URL. I was hoping to find some explanation of their calculations with regress8.

Author's Response to Decision Letter for (RSPB-2020-0142.R0)

See Appendix B.

RSPB-2020-0763.R0

Review form: Reviewer 4 (Madeleine Geiger)

Recommendation

Accept with minor revision (please list in comments)

Scientific importance: Is the manuscript an original and important contribution to its field?

Excellent

General interest: Is the paper of sufficient general interest?

Excellent

Quality of the paper: Is the overall quality of the paper suitable?

Excellent

Is the length of the paper justified?

Yes

Should the paper be seen by a specialist statistical reviewer?

No

Do you have any concerns about statistical analyses in this paper? If so, please specify them explicitly in your report.

No

It is a condition of publication that authors make their supporting data, code and materials available - either as supplementary material or hosted in an external repository. Please rate, if applicable, the supporting data on the following criteria.

Is it accessible?

N/A

Is it clear?

N/A

Is it adequate?

N/A

Do you have any ethical concerns with this paper?

No

Comments to the Author

I was a reviewer of a previous version of this manuscript and would like to thank the authors for taking almost all of my suggestions into account (or discussing their contrasting view satisfactorily). I am particularly happy that my comment on sexual dimorphism helped to expand the analyses, which resulted in highly interesting results in line with the domestication hypothesis. This whole topic is very exciting and I'm looking forward to this and further studies about parallels between domestication and urban evolution. After all, human-animal interactions are continuous and some of the basic environmental aspects during initial phases of domestication in e.g. dogs and cats were probably similar to the conditions in which our urban foxes and other urban animals are living today (e.g. Vigne 2011, *The origins of animal domestication and husbandry: A major change in the history of humanity and the biosphere*; Zeder 2012, *Pathways to Animal Domestication*)

I have only relatively minor comments to add, as outlined below:

I think that the reference list does not match the citation-numbers in the text in many cases. Could it be that the newly suggested papers (e.g. Wandeler et al. 2003, Gloor et al. 2001) are not yet added to the reference list, which resulted in a shift?

Speaking of which: There is actually also a more recent paper, similar to the Wandeler et al. 2003 paper, which is re-examining the same topic. This might also be worth citing, although I think you already do, despite the reference not being shown in the list: DeCandia et al. 2019. *Urban colonization through multiple genetic lenses: The city-fox phenomenon revisited. Ecology and evolution, 9(4), 2046-2060.*

Typo on L. 67: point instead of comma between citations.

L. 110. Wouldn't it be the other way around? Larger brain and hence larger braincase?

L. 150 and also in other instances: Clade names above the genus level are normally not in italics. Sorry for being picky.

Fig. 2 caption: Thank you for changing the group "colours" to greyscale, as suggested in my previous review. I think the figure caption should be adapted accordingly.

L. 270. You write that "...the dorsal aspect being reduced in the urban habitat". I do not quite understand what you mean here.

L. 271 ff. I find it hard to see and trace the described skull shape changes in Figure 3. Would it be possible to describe which landmarks are suggesting the described changes, so one could compare this with Fig. 1 and 3? (I know that I've asked you in the previous review to not magnify the differences, and I think your argumentation is sound. However, I would ask the same if the changes were not magnified).

L. 293 – 295. The statement about the "increased nasal region in urban foxes" kind of contradicts the statement above, which is that urban foxes have a reduced maxillary region and a shorter snout compared to the rural ones. Could you clarify, please?

L. 306 – 310. If I'm not misinterpreting your results this means that skull shape of female foxes is similar to urban foxes (of both sexes). Is this correct? Could this also be interpreted along the line that urbanisation leads to 'feminization', in accordance with what is usually suggested to be the case in domestication?

L. 321 ff. and Fig. 5. It looks a bit as if PC1 of the ventral aspect and PC1 of the dorsal aspect would depict opposite patterns (same with PC2). For example, in dorsal aspect, *V. macrotis* has a relatively short snout and narrow zygomatic arches, whereas from the ventral aspect, *V. macrotis* has a relatively long snout and wide zygomatic arches. Maybe I am totally not getting it here. However, could you elaborate on that, please?

L. 399. An additional, interesting line of investigation would also be if there is an increase in the occurrence of aberrant coat colour in urban foxes (e.g. white spots), such as seen in a wide variety of domestic animals, farm foxes, and also these wild mice selected for tameness I was mentioning in the first review.

Lastly, I would like to shortly go into our discussion on allometry: I was asking if variations of skull shape between urban and rural (or male and female foxes) might be because of allometry. You were answering that getting rid of the effect of body size would potentially omit also biologically relevant information. I completely agree with this and your approach. I just want to point out that I did not talk about "ontogenetic" allometry (as you said, this is avoided via only using adults), but about "static" allometry. This is: skull shape varies because of differences in body size among the adult individuals of a species/population. This is an important pattern influencing adult skull shape e.g. in domestic dogs: small breeds have proportionally larger brain cases and shorter snouts as adults than larger breeds. In the cases of foxes I was thinking, if e.g., the adult urban foxes are smaller or larger than the adult rural foxes, parts of the observed skull shape variation might be explained by these size differences. The same argumentation would apply to the sexes. I'm sorry if I'm stating the obvious here. I just think it would be worth mentioning this in your paper. It would of course be great if you could check if urban and rural populations are indeed different in size, or not (e.g. comparison of absolute distances between certain landmarks as a proxy for body size or also via centroid size). However, this is most probably too much asked at this point.

Decision letter (RSPB-2020-0763.R0)

27-Apr-2020

Dear Dr Parsons

I am pleased to inform you that your manuscript RSPB-2020-0763 entitled "Skull morphology diverges between urban and rural populations of red foxes mirroring patterns of domestication and macroevolution" has been accepted for publication in *Proceedings B*. Congratulations!!

The referee(s) have recommended publication, but also suggest some minor revisions to your manuscript. Therefore, I invite you to respond to the referee(s)' comments and revise your manuscript. Because the schedule for publication is very tight, it is a condition of publication that you submit the revised version of your manuscript within 7 days. If you do not think you will be able to meet this date please let us know.

Sincerely,

Dr John Hutchinson, Editor

Associate Editor

Comments to Author:

As mentioned previously, I think this paper deals with topical subject, and the authors have accumulated a very nice anatomical data set with which to address the questions surrounding urbanisation on animal form-function. The reviewers, on the whole, have shared this positive outlook throughout, on both the topic and the attempt to analyse the data. The current reviewer suggests some reasonable minor improvements to improve clarity and smooth out the rough edges on certain aspects of the paper, and I'm inclined to agree that these should be addressed before publication. It is my recommendation that the authors be given the opportunity to resubmit a modified version to incorporate the minor changes suggested by the reviewer.

Reviewer(s)' Comments to Author:

Referee: 4

Comments to the Author(s).

I was a reviewer of a previous version of this manuscript and would like to thank the authors for taking almost all of my suggestions into account (or discussing their contrasting view satisfactorily). I am particularly happy that my comment on sexual dimorphism helped to expand

the analyses, which resulted in highly interesting results in line with the domestication hypothesis. This whole topic is very exciting and I'm looking forward to this and further studies about parallels between domestication and urban evolution. After all, human-animal interactions are continuous and some of the basic environmental aspects during initial phases of domestication in e.g. dogs and cats were probably similar to the conditions in which our urban foxes and other urban animals are living today (e.g. Vigne 2011, *The origins of animal domestication and husbandry: A major change in the history of humanity and the biosphere*; Zeder 2012, *Pathways to Animal Domestication*)

I have only relatively minor comments to add, as outlined below:

I think that the reference list does not match the citation-numbers in the text in many cases. Could it be that the newly suggested papers (e.g. Wandeler et al. 2003, Gloor et al. 2001) are not yet added to the reference list, which resulted in a shift?

Speaking of which: There is actually also a more recent paper, similar to the Wandeler et al. 2003 paper, which is re-examining the same topic. This might also be worth citing, although I think you already do, despite the reference not being shown in the list: DeCandia et al. 2019. *Urban colonization through multiple genetic lenses: The city-fox phenomenon revisited. Ecology and evolution*, 9(4), 2046-2060.

Typo on L. 67: point instead of comma between citations.

L. 110. Wouldn't it be the other way around? Larger brain and hence larger braincase?

L. 150 and also in other instances: Clade names above the genus level are normally not in italics. Sorry for being picky.

Fig. 2 caption: Thank you for changing the group "colours" to greyscale, as suggested in my previous review. I think the figure caption should be adapted accordingly.

L. 270. You write that "...the dorsal aspect being reduced in the urban habitat". I do not quite understand what you mean here.

L. 271 ff. I find it hard to see and trace the described skull shape changes in Figure 3. Would it be possible to describe which landmarks are suggesting the described changes, so one could compare this with Fig. 1 and 3? (I know that I've asked you in the previous review to not magnify the differences, and I think your argumentation is sound. However, I would ask the same if the changes were not magnified).

L. 293 - 295. The statement about the "increased nasal region in urban foxes" kind of contradicts the statement above, which is that urban foxes have a reduced maxillary region and a shorter snout compared to the rural ones. Could you clarify, please?

L. 306 - 310. If I'm not misinterpreting your results this means that skull shape of female foxes is similar to urban foxes (of both sexes). Is this correct? Could this also be interpreted along the line that urbanisation leads to 'feminization', in accordance with what is usually suggested to be the case in domestication?

L. 321 ff. and Fig. 5. It looks a bit as if PC1 of the ventral aspect and PC1 of the dorsal aspect would depict opposite patterns (same with PC2). For example, in dorsal aspect, *V. macrotis* has a relatively short snout and narrow zygomatic arches, whereas from the ventral aspect, *V. macrotis* has a relatively long snout and wide zygomatic arches. Maybe I am totally not getting it here. However, could you elaborate on that, please?

L. 399. An additional, interesting line of investigation would also be if there is an increase in the occurrence of aberrant coat colour in urban foxes (e.g. white spots), such as seen in a wide variety of domestic animals, farm foxes, and also these wild mice selected for tameness I was mentioning in the first review.

Lastly, I would like to shortly go into our discussion on allometry: I was asking if variations of skull shape between urban and rural (or male and female foxes) might be because of allometry. You were answering that getting rid of the effect of body size would potentially omit also biologically relevant information. I completely agree with this and your approach. I just want to point out that I did not talk about "ontogenetic" allometry (as you said, this is avoided via only using adults), but about "static" allometry. This is: skull shape varies because of differences in body size among the adult individuals of a species/population. This is an important pattern influencing adult skull shape e.g. in domestic dogs: small breeds have proportionally larger brain cases and shorter snouts as adults than larger breeds. In the cases of foxes I was thinking, if e.g., the adult urban foxes are smaller or larger than the adult rural foxes, parts of the observed skull shape variation might be explained by these size differences. The same argumentation would apply to the sexes. I'm sorry if I'm stating the obvious here. I just think it would be worth mentioning this in your paper. It would of course be great if you could check if urban and rural populations are indeed different in size, or not (e.g. comparison of absolute distances between certain landmarks as a proxy for body size or also via centroid size). However, this is most probably too much asked at this point.

Author's Response to Decision Letter for (RSPB-2020-0763.R0)

See Appendix C.

Decision letter (RSPB-2020-0763.R1)

05-May-2020

Dear Dr Parsons

I am pleased to inform you that your manuscript entitled "Skull morphology diverges between urban and rural populations of red foxes mirroring patterns of domestication and macroevolution" has been accepted for publication in Proceedings B.

Open Access

You are invited to opt for Open Access, making your work freely available to all as soon as it is ready for publication under a CC BY licence. Our article processing charge for Open Access is £1700.

Paper charges

Sincerely,

Appendix A

Associate Editor

Board Member: 1

Comments to Author:

Thank you for the opportunity to review this paper. The impacts of humanisation of the environment on macro-level anatomy and functional morphology is a highly topical subject, and I found this analysis of skull shape in foxes (using a very nice anatomical data set) to be very interesting. Firstly, I would like to apologise for the extremely lengthy delay with peer review. As the authors will see, reviewers 1 and 2 provided very different views of the manuscript and this necessitated a third review, which unfortunately took a long time to find. Based on my own reading of the manuscript, and the views of the reviewers, it would be my recommendation that the paper be rejected in its current form, but that the authors have the opportunity to revise and resubmit to Proceedings B. In particular I think the majority of points raised by reviewers 1 and 3 in their comments require careful consideration.

Thank you for this advice. We have gone through the reviews and have focused on addressing the concerns of reviewers 1 and 3 as suggested. We have also addressed a number of issues raised by reviewer 2 where we saw they would improve the manuscript.

Reviewer(s)' Comments to Author:

Referee: 1

Comments to the Author(s)

This is a very interesting manuscript looking at the morphological differences between urban and rural foxes that shows some intriguing parallels between urbanisation and domestication/ macroevolutionary trends. This is a really valuable dataset and I am surprised it hasn't been studied more since its collection in the 1970s. On the whole, I think the manuscript is presented and written well. The context given in the introduction seems appropriate and the methods, as far as I can tell, have been carried out correctly. For the most part, the results are presented well and the interpretation of those results is justified. I just have a few comments:

Line 16: Add 'be' after 'found to'

Line 35: 'cities should' feels like odd phrasing to me. Maybe 'cities may'?

Line 112: It would be good to see a table of specimens, with sex and habitat (and location/grid reference) in the supplementary info.

We have made these suggested changes and now include the suggested information for specimens in a supplementary table.

Line 114: What were the criteria for classifying specimens as urban or rural?

We thank the reviewer for pointing this out. We have now expanded this description in our text and as mentioned above have included coordinates and some description for the collection sites in an appendix. We also concede there are some limitations to our approach

(due to us not collecting samples ourselves) relative to some recent developments in how to assess the 'urban-ness' of environments (see lines 118-124) , but also cite a recent article from Proc B that uses a similar approach to designate urban and rural populations of mammals (Snell-Rood and Wick 2013). Nonetheless, while we are limited by the availability of collections, recent advances could serve as a good model for new collections of foxes going forward.

Line 123: Why did you choose to analyse both dorsal and ventral views? I'm not sure that they provide different information from one another. Why was a lateral view not included? This could have provided useful information on cranial and rostral height. Indeed, why not use 3D data – was it simply a lack of access to 3D digitising equipment? In your discussion, I think you need to at least acknowledge the limitations of 2D data and how it might have affected your results.

Yes, as the reviewer suggests 3d would have been nice. Getting a lateral view was tested but it was challenging to keep the plane consistent across specimens. The dorsal and ventral aspects we took did not have this issue and were highly amenable to 2d morphometrics and also allowed for comparison with previous research on dogs and other canids that quantified shape variation (e.g., Drake and Klingenberg 2010). However, there was indeed a lack of access to 3d scanning equipment at the time but it is hoped we can build this system toward more comprehensive assessments of morphology in the future. Surprisingly, comparisons of power between 2D and 3D data sets reveal that there is surprisingly little gain from 3D (Jamniczky et al. 2015, *Evol Biol* 42:260–271 Navarro and Maga 2016, *g3* 6: 1153-1163; McWhinnie and Parsons 2019 *Env. Biol. Fish.* 102: 927-938).

We now address this limitation in our discussion on lines 355-357.

Line 218: I appreciate that the MANOVAs on partial warp scores have indicated significant differences between habitats and sexes, but I would have liked to see the distribution of the data across a morphospace to get an idea of how clearly split the groups are.

We have now added a frequency histogram for the DFAs derived from our analysis using the candisc package (see Figure 2). This plots the canonical root scores related to habitat divergence while controlling for the effects of sexual dimorphism. This is appropriate in this case simply because a PCA approach would not parse variation on the basis of habitat like a DFA/MANOVA approach does.

Line 227: Clarify the way in which the sagittal crest has been extended – anteriorly or posteriorly?

Posteriorly, thanks

Line 230: change 'could' to 'would'

done

Line 230: 'a greater need for muscle attachment' is a slightly odd construction. 'could indicate an increased area of attachment of the temporalis' is what you mean I think.

This has now been changed for greater clarity

Line 238: Harder foods do not need a 'higher mechanical advantage to access' – they simply require greater force.

We agree and have changed this statement (now appearing on line 284)

Line 238: An 'expanded sagittal crest' would not necessarily increase mechanical advantage of the temporalis.

Changed to remove mechanical advantage

Line 239: What do you mean by an expanded posterior region of the zygomatic arch? Are you referring to a greater width of the zygomatic process of the squamosal? And how does this fit with your statement earlier that urban foxes have a reduced zygomatic region (line 228)?

We meant the squamous temporal are expanded in urban foxes, it is the case that the zygomatic arch is reduced. We have changed this text.

Line 241: Actually Santana and Dumont are somewhat cautious about the supposedly high stresses in the zygomatic region in their bat FE models, suggesting they might be modelling artifacts. A better reference might be Wroe et al (2007; Proc B 274: 2819-2828) who actually look at the masticatory biomechanics of a canid (the dingo).

Thank you for the clarification, we have change the reference to reflect the more biologically appropriate comparison that Wroe et al. make (i.e., Dingo vs. bat).

Line 250: In what way are the zygomatic region and the cranium reduced in females and males respectively?

The region is larger in part because the squamous temporalis protrude more in males and create a larger space between the zygomatic arch and the frontal bone. The text has changed to reflect this.

Line 262: I was pleased to see the morphospaces in Figure 4, but I think you need to describe the results of the PCA – how are the species distributed, what is the shape change along the axes, where do red foxes plot compared to other Vulpes species?

We have now described the PCA results in terms of what happened on PC1, and where red foxes lie in that morphospace (lines 270-274).

Line 262: Why was only PC1 tested for phylogenetic signal? Blomberg's K can be calculated for multivariate data and the Vulpes specimens seem to spread across PC2 mostly.

Thank you for the suggestion. We have added a multivariate assessment of Blomberg's K to our analyses that includes PCs 1-6. We find that our data still lacks phylogenetic signal. Based on another reviewer's suggestion we also assessed the power of our K analysis and found ample power to detect an effect despite low sample sizes.

Line 271: Add 'of' after 'pattern' - **done**

Supplemental table 1: Why are the species in this order? It is neither alphabetical nor phylogenetic as far as I can see. Also, *Vulpes rueppellii* needs to be italicised.

Supplemental table 2: 'palantine' should be 'palatine' in ventral landmarks 8, 10 and 12.

Referee: 2

Comments to the Author(s)

Line 49, refs 5-6: This statement and the references are correct, but it should be noted here that recent work has questioned the universality of the syndrome, and this is important as it should lead to qualifications and caveats when making generalisations. Still, the patterns in CANIDS, to which foxes belong, is there, so this is not fatal to the ideas presented here, but it is as stated a misrepresentation of the current knowledge not to refer to this work.

Sánchez-Villagra MR, Geiger M, Schneider RA. 2016. Taming the neural crest: A developmental perspective on the origins of morphological covariation in domesticated mammals. *Royal Society Open Science* 3 160107.

We thank the reviewer for this comment now cite this paper here (line 50)

Concerning the statements in lines 60-61 on modularity favouring change in snout independent of the rest and so on, this statement should be qualified in view of both Drake and Klingenberg 2010 and Curth et al 2017 (ref below), which show that the degree of modularity in dogs is no different from that of wolves or other carnivorans, and no association with disparity was found. It is still possible and reasonable to state that modularity facilitate what is stated there, but this is not something peculiar to dogs – this should be made clear, because it weakens the argument of dev bias of line 58 as presented.

Curth, Stefan & S. Fischer, Martin & Kupczik, Kornelius. (2017). Patterns of integration in the canine skull: An inside view into the relationship of the skull modules of domestic dogs and wolves. *Zoology*. 125. 10.1016/j.zool.2017.06.002.

Thank you for this important point, we now address this and cite this paper.

Line 70 at the end: 'changes' – do you mean morphological changes? If so, please state that. From what follows it seems you mean exclusively skull changes. I would be specific there

then.

In the paragraph starting line 69, I would add the neural crest hypothesis as a third alternative, and present the three hypothesis as not exclusive from each other.

We agree that the neural crest hypothesis could link both of the other hypotheses. At this point in the manuscript we aimed to keep things phenotypic rather than mechanistic as we felt the NCC hypothesis would be less familiar to a general audience. We feel the links between these hypotheses are mechanistic and make this clear in the discussion.

Line 78: I understand the logic of using foxes to discern between the two hypothesis – but it is a little of a stretch, and if we consider the third, alternative hypothesis (neural crest), also considered by the authors, the logic here is not so strong. This should be acknowledged.

We don't feel the NCC hypothesis is competing, rather it is a mechanistic hypothesis that potentially links everything.

Line 81: 'morphological divergence' – this refers exclusively to skull shape – it would be fair to be explicit about that from the start. This is perfectly fine, but it should be made clear.

Line 97: braincase, one word

We have fixed this typo, thanks

Line 98: for the 'increased cognitive demands' leading to 'larger braincase' reference 1 is provided, which is on butterflies – please change that. Actually, citing the works of Kruska and other here would not fully fix this idea, as it is contested this relation when it comes to subtle differences and here it would be more relevant to differentiate among different parts of the brain.

Lines 125-126: unclear what is meant here with 'basal species'

The various lineages of canids are well worked out, therefore we know which species are more anciently derived relative to Vulpes, hence we call them basal. We now clarify this text and cite the phylogeny of canids here to help the audience.

Line 128: species of what?

fixed

Lines 171-176 and what follows:

I understand how having a baseline of comparison using a phylogenetic bracket is useful – so the changes in the foxes can be put in context by comparing with several Carnivora, 'correcting' or considering phylogeny. But what we have here are samples of foxes in cities versus rural habitats from 1972-1973 and then samples of several species of Carnivora - so

Commented [AJC1]: Include? I think that this brings us into a realm that we don't want to be in. I feel like I could try to argue a case for mechanisms being important – I think if we drop this sentence the reviewers accept our reasoning in the first sentence without issue.

the statement in lines 175-176 'To assess whether divergence across the urban/rural habitat axis was influenced by phylogenetic effects...' makes not much sense to me. Maybe it is the way it is expressed. How could phylogeny influence such as differentiation? Surely phylogeny influences everything and we need to account for it, but here some fancy analysis does not fit I would say, at least as it is presented.

This is correct, we have adjusted this text to help clarify our meaning. We simply meant to determine whether patterns of divergence at both levels were similar.

I do not see then the significance of the lines 268-270 to start with.

we are not clear on what is meant by this comment as these deal with the sexual dimorphism we found

I recommend to add relevant commas in sentences in the following lines: 224, 251, 257, 259.
done

I evaluate the discussion in page 10, lines 235-247 as original, as these functional aspects have not been discussed in the recent literature dealing with changes in skull shape driven by domestication (eg Evin and colleagues work on pigs, Sánchez-Villagra et al. work on domestics skull growth).

Thanks, this is helpful to know

Lines 284-285: the developmental bias hypothesis here is not clear.

We now expand on this here

Line 297: Wilkins, not Wilkens. See other recommended literature here. Lines 303-304: there is also a recent review by Wilkins AS 2017 REVISITING TWO HYPOTHESES ON THE DOMESTICATION SYNDROME" IN LIGHT OF GENOMIC DATA (Russian Journal, in EN) which is quite relevant here.

fixed, and citation added

This is a recent and relevant paper discussing the parallels with island and domestication, neural crest hypothesis, etc., treated in this manuscript.

van der Geer AAE. 2019. Effect of isolation on coat colour polymorphism of Polynesian rats in Island Southeast Asia and the Pacific. PeerJ 7:e6894 <https://doi.org/10.7717/peerj.6894>

thanks for this info, we will keep it in mind for future research

Referee: 3

Comments to the Author(s)

Main problems

1. Absence of evidence is not evidence of absence. Would their value of K, if the parameter

equalled the estimate, put an important phylogenetical spoke into the wheels of the later analysis? (p11 lines 263-265 suggest rather little power).

We disagree with the reviewer here. We don't have support for the alternative hypothesis that there is phylogenetic signal, instead we have evidence in support of the null (no phylogenetic signal). The tests performed account for this and were done with proper levels of statistical power.

We are unsure which of our later analyses the reviewer is concerned with. The phylogenetic tests are intended to account for a possible influence on divergence in *Vulpes*. Tests of divergence between urban and rural foxes should not directly include phylogeny as a factor- this is not of concern in cases of microevolution, nor is it possible because there is no known phylogenetic separation between urban and rural red foxes. In this sense there would be no 'spoke' into the wheels since our intention was to only provide possible explanations for the patterns of divergence we found at the microevolutionary level. By taking the steps to examine patterns of shape variation across the genus we find it very interesting that some of the pattern of divergence in the urban/rural gradient appear qualitatively and quantitatively similar to that seen across the genus.

With regards to power we have now provided details of a power analysis which showed that we had adequate power to detect phylogenetic signal if it was present (lines 319-324).

Specifically,

(a) the authors more than once interpret absence of evidence as evidence of absence. Apart from justifying their later analyses, the passage in lines 307-309 is a clear example.

(b) they rely on no phylogenetic signal for a whole set of analyses

We have responded to comment (a) above. Yes, there would be no need to account for phylogenetic signal in the microevolution between urban and rural foxes since this divergence has occurred over the past 70 years. However, knowing the patterns of divergence across the genus, and within the urban/rural habitat gradient was done because they can be highly complementary. We found support in favour of the null hypothesis that there was no phylogenetic signal which our new analysis reveals there was adequate power to test. We also found evidence in favour of the null hypothesis that trajectories of urban and rural divergence and macroevolution were statistically aligned. Within the main text we now explicitly state that our findings are in support of a null hypothesis.

(c) it seems likely they have little power in estimating K

I note that quite a lot of the paper seems to depend on these analyses and doubtful interpretations.

In response to this we have now explicitly tested the power of our analyses. We found our results from the multivariate assessment of Blomberg's K were robust to low sample size and exhibited high levels of power. As mentioned above these results are now provided from lines 319-324.

2. Misinterpretation of confidence of intervals

p11 line 275. The main cause of width of confidence is inadequacy of data (not enough, not informative enough about the parameter). To interpret confidence intervals as though they reflect the world, rather than the imperfections of our perceptions of it, is to try to make a silk purse out of a sow's ear. (In fact the statement is ambiguous about the comparison being made between the confidence intervals. Is it (i) in position, so one CI is somewhat to the right of another CI (ii) in position, but one CI and the other don't overlap (this would usually be recordable in a significance test), or (iii) in width, so one is wider than the other, without regard to position?)

We disagree that we have misinterpreted our analysis based on confidence intervals. The width of a confidence interval can indeed be due to inadequacy of data as the reviewer asserts, but it can also be due to real variation in the data. In line with the we have a much larger sample size for the urban/rural foxes (n=111,). Yet, the confidence intervals within these foxes is wider (177°, 108°) than that found within the phylogenetic sampling (n=12 species mean shapes, CIs= 48°, 32°). So, it seems we do not have an issue with inadequate data here. Nonetheless, we have extended this analysis with an alternative modelling approach through the RRPP package in R. The linear models in this approach were able to distinguish whether the distance (i.e. magnitude of morphospace covered) differed between macro and microevolutionary divergence, while also providing a different type of test for differences in the angle of divergence. Again, we found support in favour of the null hypothesis that angles did not differ, but we did find evidence that the magnitude of divergence differed. This is not surprising but is interesting because it suggests that while macroevolutionary divergence covers more morphospace, *the way* evolution occurs does not differ from the urban/rural divergence.

We have clarified the text in this section and have added in details of our new analysis in the methods (lines 254-261) and in the results and discussion (lines 339-342).

Minor thoughts

1. The authors don't say whether the Habit*Sex interaction is of the kind that indicates a different relationship e.g. multiplicativity, which might in principle be abolished by a transformation; or a qualitative difference in how the sexes respond to habitats.

We are unsure of the issue being raised here, nor to our knowledge is it standard to report such details in MANOVA results. This is definitely not a qualitative difference as we test this with MANOVA. We could use clarification from the reviewer but this interaction could be due to differences between sexes in the magnitude or direction (or both) to the urban/rural gradient. Deriving this could be possible but we don't see how this adds to the story of the paper especially given limited space and this is indeed a 'minor thought' (we would have to

explain more methods, results, and interpretation).

2. Family group problems are possible, causing non-independence of the samples, depending on how the skulls were collected. I haven't looked at any of the referenced papers, but there is the possibility that there are sampling biases of which it would be helpful to be aware.

We now address this limitation (line 116)

2a Pursuing that thought, the urban/rural divide will in fact be a number of different urban places and a number of different rural places. Most obviously, the urban will divide by conurbation. In principle, we could regard these conurbations as random effects. Alternatively, a more granular analysis could show that the same pattern occurs in the different conurbations (perhaps by absence of interactions with conurbation in an ANOVA or MANOVA). However, most "accidental sampling" studies will have potential problems of this kind, and I don't want to over-stress them.

Yes, we feel this sampling makes our result more robust in that sites are likely to vary in their habitat variables, yet we get a clear pattern of divergence between urban/rural sites. In future studies it would be ideal to collect contemporary samples and be able to provide much more detail on how we defined urban/rural. As is the case these samples were simply collected well before the ideas of urban ecology or contemporary evolution were likely even conceived.

3. In the MANOVA tables on p19, p-values are given with a > instead a <, Thus, for example, ">0.001" should read "<0.001".

This is now fixed

4. The sample sizes of the species (p25) are very uneven, making the species-level analyses not so persuasive (as the balance of within-species and between-species causes of variation in the mean will be different for the different species, making homoscedasticity assumptions less credible).

As stated in the original manuscript we used the mean shape for each species in our analyses, not the full sample size. We never modelled the full sample size so this minimizes issues with uneven sampling that was a constraint on our study that arose from the museum collections that were available to us. We now highlight this step on line 169.

5. Would it not be natural to control for species in the MANOVAs on p19?

This would simply not be possible because the MANOVAs are performed on one species- the red fox. The MANOVAs did not involve different species, and simply can't because habitat

(urban/rural) is not a factor we have information on for all of the other species, nor is it likely relevant for most of them.

6. In the MANOVA on p19, both interactions are given p-values of 0.05. Looking them up myself from F, DF1 and DF2, they are both just less than 0.05. I see the value in not giving too many decimal places for approximate values (as the F is approximate), but on the other hand, the reader should somehow be informed that the exact lookup of the approximate result is actually less than 0.05 rather than, say, 0.054. It is one source of p-value-inflation that need not be worried about.

We previously followed the standard practise of rounding numbers up but now include an extra decimal place for these p-values.

Appendix B

Associate Editor Board Member: 1

Comments to Author:

As mentioned previously, I think this paper deals with topical subject, and the authors have accumulated a very nice anatomical data set with which to address the questions surrounding urbanisation on animal form-function. The reviewers generally share this positive outlook on both the topic and the attempt to analyse the data. However, reviewer 2 suggests some reasonable improvements to improve clarity and smooth out the rough edges on certain interpretations. Reviewer 3 remains concerned about some of the statistical analyses and particularly the way they are rationalised and presented. It is my recommendation that the authors be given the opportunity to resubmit a modified version to incorporate the changes suggested by Reviewers 2 & 3.

Dear Editor,

We thank you for these latest reviews and have responded to them below. We are pleased that reviewer 1 sees no further need for change, and that reviewer 4 has provided a number of helpful comments and insights that will improve the manuscript. We have taken most of these points on board as you will see below. However, we are unclear why reviewer 3 is questioning our statistical methods, given that these are widely used and for which we have provided numerous references. We respond below to reviewer 3's comments as best we can but do not feel it is necessary to change our analyses given their widespread usage.

Reviewer(s)' Comments to Author:

Referee: 1

Comments to the Author(s).

I am satisfied that the authors have addressed all my comments appropriately. As far as I can tell they have also addressed the comments of the other reviewers in a sensible fashion.

We thank the reviewer for this opinion

Referee: 4

Comments to the Author(s).

This manuscript is about cranial shape variation in urban vs. rural red foxes. The authors found that there are significant differences in skull shape between urban and rural foxes and that these shape changes are consistent with the ones found in an interspecific comparison among other *Vulpini*. This is an interesting study on the subject of urban adaptation, which is still far to less extensively investigated, despite its rising importance in a steadily urbanizing world.

The data set is exceptionally extensive and, as far as I can judge, the applied methods are sound. However, I feel that the manuscript lacks clarity in many instances (see my detailed comments below). The terminology and the descriptions related to taxonomy, systematics, and concepts of evolution, development, domestication, and breed formation are oftentimes quite peculiar and unusual compared to subject-specific literature, as well as too simplified, in some parts at the brink to non-correct (again, see my detailed comments below). Further, I would strongly recommend to include further references in various instances (also as indicated below) and to extend the discussion. For example, the authors discuss only adaptive concepts, without taking into account non-adaptive processes, which might explain their findings.

We thank the reviewer for this constructive overview. We feel we have an especially powerful dataset that enables us to address a broad set of topics. Most of the comments provided here have now been taken on board unless stated otherwise. Several references have been added and sections clarified as suggested.

Detailed suggestions for edits:

L. 15: Genus name *Vulpes* in upper case.

Changed throughout

L. 20-21: I think your sampling of the genus *Vulpes* is actually not complete, as stated here. According to the standard reference work by Wilson & Reeder (Mammal species of the world, 2005, 3rd edition), there is also *Vulpes pallida*, which you did not sample. Please rephrase, or specify later which reference for *Vulpes* you were using.

Thanks, we have rephrased this by replacing 'complete' with 'extensive' which is more accurate (line 21)

L. 23 & L. 40: Do you mean divergence "of skull shape" between urban and rural habitats"? Please make clear what you mean. Otherwise it sounds as if the habitats themselves would diverge.

Thanks, we have changed these lines to clarify that we mean phenotypes on line 41

L. 24-25: This sentence basically repeats the previous one. Maybe delete?

We have now edited this sentence.

L. 25-26: Why "additionally"? Dogs are one example of many where this shortening of the

snout has occurred during domestication. Plus, please be aware that domestication is not equal to breed formation, so the term “breed” should not be used here.

We have removed dogs as an example. We do however think they are of particular interest in relation to foxes because they are both canids. Therefore, we feel that our data set could lend itself especially well toward hypotheses explaining the initial steps in the domestication of wolves. Line 27

L. 37-40: Maybe add here, that selection could not only be different across a suite of functionally salient traits, but also across clades.

We are unsure what the reviewer means here by clades. However, in relation to this we have added in the mention that we do not know whether this divergence is present in replicate sites (i.e. other cities and adjacent rural sites). Line 44

L. 48-49: It would be appropriate to cite also the original paper by the people who came up with the term ‘domestication syndrome’: Wilkins et al. 2014 (I think it’s your reference 44).

Thanks, we have cited this now.

L. 48 & 51: What do you mean by “stereotypical”? That similar changes occur in different domestic species? Please clarify.

We now specify different species

L. 50-53: This list is not exhaustive and I would recommend to include “for example” or something similar.

Done

Paragraph on domestication syndrome: I would strongly recommend to also cite the new paper by Lord et al. 2019, who critically discuss the ‘domestication syndrome’: Lord, K. A., Larson, G., Coppinger, R. P., & Karlsson, E. K. (2019). The history of farm foxes undermines the animal domestication syndrome. *Trends in Ecology & Evolution*.

Very interesting paper! We now cite this, and it is an important caveat that some amount of domestication in this famous experiment could have been pre-dated by the source of foxes used. Line 80

Further, and I have already mentioned this above: the selection for a shorter snout in some dog breeds (as described e.g. by Drake & Klingenberg 2010) has nothing to do with the domestication syndrome, but with breed formation. This means, with selection for specific traits subsequent to domestication. Domestication and breed formation are two very different processes and should not be mixed.

Fine comment, we have clarified this distinction in the text. However, we do respectfully disagree that they have nothing to do with each other as developmental bias may unify both processes.

L. 55: There is no evidence that the extensive morphological change in dogs has been particularly fast. See e.g. Geiger, M., & Sánchez-Villagra, M. R. (2018). Similar rates of morphological evolution in domesticated and wild pigs and dogs. *Frontiers in zoology*, 15(1), 23.

We remove the use of 'rapid' here, but it does seem that both lineages (pigs and dogs) have undergone rapid change relative to most mammals we are aware of.

I would further strongly recommend to refer to the domestic form of dogs as 'domestic dogs' or the like, not just 'dogs'. This could be confused with you referring to the clade Canidae, which is usually also referred to as 'dogs'.

We have changed this throughout the text

L.58: Developmental biases 'within' dog skulls is a phrasing I've never heard before. I would delete this.

Additionally: I think artificial selection for these traits and developmental biases are not mutually exclusive. It may be that head shape development is biased in one way or the other, leading to a limited number of possible morphotypes within an evolutionary lineage. However, such morphotypes, e.g. as seen in bulldog or greyhound, would not be prevalent in these breeds if it wasn't for artificial selection, i.e., people who were selecting for these traits as they occurred.

We have removed 'within'. We agree with this opinion and it was our intention to get this point across in the text. Therefore, we have edited to enhance this point.

L. 61: Developmental modularity does not 'favour' changes in the snout independent of the rest of the skull, but it 'makes possible' (or something like this). These are developmental patterns, not "conscious" processes.

Agreed and changed to reflect this.

L. 65-66: '...shows a similar trend in skull shape.' What exactly? Skull shape change? Please specify.

We have collapsed the last two sentences in the paragraph to accommodate this point

L. 67: What do you mean by "mimic ancestral condition for dogs?" It is not clear which condition you mean. I would suggest to put write something in the line of "mimic the selection regime during the initial domestication process of dogs".

Changed in accordance with this suggestion on line 94

You could also cite here this study on mice, in which selection for tameness was more similar to the scenario in dogs and other commensal domesticates (and has led to a shortening of the head): Geiger, M., Sánchez-Villagra, M. R., & Lindholm, A. K. (2018). A longitudinal study of phenotypic changes in early domestication of house mice. *Royal Society open science*, 5(3), 172099.

Interesting paper, we are focused on canids in this section so felt it was a bit awkward to include this without adding substantial length (we are already pushing the limits for Proc b)

L. 69: "during" instead of "for"?

Done

L. 70: "driver of initial changes" of what? Please specify.

Toward domestication now added

L. 78: Why do you point out that foxes are non-social? Does this have an implication for this study? In fact, I would not regard foxes as non-social per-se, as they are sometimes living in family groups and are showing complex social behaviour. I agree that the behaviour may be less complex than in wolves (e.g., solitary vs. pack hunting), but still not non-social.

We concede that foxes have some level of social behaviour. Foxes live in monogamous pairs but not social groups such as feral dogs and wolves. It is our view that if social behaviour is seen as a driver of domestication in dogs, it is also confounded with other phenotypes. With foxes we see a much less social animal, yet a similar trajectory of morphological change (i.e. morphology is no longer confounded from the exceptionally social behaviours of dogs).

L. 84: City foxes are not only known from Great Britain, but also from Central Europe: Gloor, S., Bontadina, F., Hegglin, D., & Deplazes 2001. The rise of urban fox populations in Switzerland. *Mammalian Biology*, 66, 155-164.

Thanks, yes that is noteworthy. We cite this a bit later in the manuscript now on line 104

L. 89: In this study on urban and rural foxes in Switzerland, it has been shown that urban foxes are indeed somewhat isolated from their rural conspecifics: Wandeler, P., Funk, S. M., Lurgiader, C. R., Gloor, S., & Breitenmoser, U. (2003). The city-fox phenomenon: Genetic consequences of a recent colonization of urban habitat. *Molecular Ecology*, 12(3), 647-656.

Very useful reference, we have included this now on line 104

L.94ff: You predict that rostral shape in urban foxes would follow the line of the domestication syndrome, but for brain size and sexual dimorphism it would be the opposite. This is contradictory. After all, in urban house finches, it was found that brain and eye size is smaller compared to rural conspecifics (Hutton & McGraw 2016). Further, in general, domestication is associated with a decrease of sexual dimorphism.

Hutton, P., & McGraw, K. J. (2016). Urban–rural differences in eye, bill, and skull allometry in house finches (*Haemorrhous mexicanus*). *Integrative and comparative biology*, 56(6), 1215-1224.

Thanks for this input, we now point out that our brain size prediction contradicts domestication syndrome (line 113) but to remain objective we leave the sexual dimorphism hypothesis as is (even though shape differences indicate it may be smaller in the urban habitat).

Our sexual dimorphism hypothesis was directed at differential effects of habitat on sexes, but was not directional whereby we would expect sexual dimorphism to decrease in the urban habitat. So, this raised an interesting question and we have now performed an additional analysis to address this directional prediction. Interestingly we now find support for this prediction where the ventral aspect trends toward reduced dimorphism in the urban habitat, and the dorsal aspect shows a significantly reduced amount of dimorphism. The methods for this are now provided, and findings are now briefly discussed. We thank the reviewer for these ideas as it strengthens support for urban habitats matching up with domestication syndrome. Methods described in Lines 177-187

Hutton and McGraw is cited on line 51 now to set up this prediction

L. 100-102: The comparison rural vs. urban is intraspecific. I do not see why these should be biased by phylogenetic relatedness and why this analysis is important here.

We feel the reviewer has misinterpreted our intentions, or misunderstands our interests. We certainly would not expect phylogenetic effects to differ between urban and rural populations. However, we did feel that phylogenetic effects within the Vulpini could be influencing the response to urban/rural habitat gradients. We have clarified this text to get the point across more explicitly. While we did not find phylogenetic effects on shape difference within Vulpini, the alignment of their divergence with urban/rural divergence is suggestive of developmental bias influencing evolution at different scales. Lines 121-125

L. 108 ff. (Materials and Methods): Would it be possible to provide the number of specimens in each habitat (rural and urban) as well as the sex in a table or written out in the text in the main manuscript, please? This would be crucial for the interpretation of the results, especially to see if there were e.g. equal numbers of males and females for each habitat. It's good to have the supplementary table with all the specimens, but a summary would also be important for the main text.

Done, although we have done this in the text to save the space needed for another table.
Line 131-133

L. 111: I would recommend to write: Which are "housed" in the collections of the...

done, line 133

L. 113: All specimens "have"
done, line 135

L. 124-125: So I guess the fusion of the basisphenoid synchondrosis is associate with age?
Could you please provide a reference here?

Done, line 147 with a new reference provided:

Roulichova J, Jana AM. 2007. Simple method of age determination in red fox, *Vulpes vulpes*.
Folia Zoologica 56, 440-444

L. 129: Maybe "aspects" instead of "views"?

Changed throughout

L. 129-131: This is a peculiar phrasing. Dorsal and ventral aspects do not "provide" landmarks.
Rather, you have defined these landmarks on these aspects. Please rephrase.

Some minor rephrasing has fixed this on lines 151-153

Figure 1:

- Delete "the" and "genus" in "the *Vulpes* genus". It's clear that *Vulpes* is a genus.
- Landmark descriptions are actually in supplementary table 3.

Done but it is supplementary table 2 where the landmarks are

Supplementary Table 3.

- Landmark 5: delete duplication of words.
- What is the difference between landmarks 2 and 3 (and 7 and 8)? Please describe 2 and 7 more accurately.

Supplementary table 2 actually, landmark 3 is at the *maxillary* suture, landmark 2 is at the anterior *tip of maxillary* suture. Landmark 5 is fixed for repetition.

L. 135: The last part of this sentence seems not to belong there. Rephrase?
done

L. 136: This is actually Supp. Table 2, not 1. I think this table, or at least the names of the species, should go in the main text.

We are unsure of what the error is here, the species sampling table is supplementary table 3 in our submission. It should remain supp. Table 3 because that is the order of topics and

where the text requires its use.

Further, what do you mean by "additional canids"? As far as I see, you have sampled almost the whole clade Vulpes (for the one exception see my comment above) as well as the basal Otocyon and Nyctereutes. This means you have basically sampled the clade Vulpini. Please make this clear, as this phylogenetic context is important to understand the implications of your analysis.

We have clarified this text and added in the species list. Lines 155,158-161

L. 162: what do you mean by "they"? Do you mean you want to test if similar shape changes as seen in interspecific comparison are also prevalent in an intraspecific (urban vs. rural) comparison?

This sentence has now been clarified, line 198

L. 165 ff.: which one is the dependent and which the independent variable? Please specify. Please write again what you mean by "shape". Partial warp scores?

We stated previously in our text that partial warp scores accounted for quantitative variation in shape, line 166 . We also stated that sex, habitat, and their interaction were explanatory variables, while shape was the response variable on lines 175,176.

Also, I do not see why you performed an ANOVA first. Aren't both of your variables continuous? You were performing a regression analysis anyway, so this tells you the same thing. Or do I miss something here?

We did not perform a normal ANOVA anywhere, but we did do a Procrustes. A standard regression approach is not appropriate to statistically test the effect of categorical explanatory variables like habitat and sex. We later use a regression to model the effect of these variables on shape to produce visual aids in the form of deformation grids (line 204), but we do not use it to test the effect of these factors.

Please rephrase: there is no such a thing as "effect of allometry on shape". It would be "effect of size on shape", which is allometry or allometric scaling.

We have edited this on line 201

L. 170: You say you did this for Vulpes species, however, in the Figure showing the results of PCA, there are also the other two non-Vulpes species of Vulpini. Please rephrase.

Rephrased to account for the two basal species. Line 208

L. 171: this is actually supplementary table 4, not 3.

Need to check

L. 181: It is not 12 Vulpes species (see above), but 12 species of Vulpini. Please change (also in remainder of the text).

Rephrased throughout

L. 197: Remove brackets around (BM). I would suggest to not abbreviate this term anyway. Every abbreviation makes a text harder to read and this specific term only occurs three times in the manuscript.

Changed throughout

L.161-161 and L.184-185: These two paragraphs have a similar introductory sentence. This is confusing and I would suggest to condensate.

We have edited these so they are more easily distinguished now

L. 209: You did not perform a 'phylogenetic analysis'. This would be the analysis of phylogenetic relationships between clades. Please rephrase.

We have made a minor adjustment of the text on line 250

L. 208-211: This is confusing. You are performing the test to determine if phylogeny has an effect, but saying here that there is no need to account for a phylogenetic signal. Maybe I do not get the point here, but I think it would be helpful to rephrase.

We have approached our data objectively but have also now edited this text in the manuscript to address these concerns on lines 249

L. 224: delete one "run"

Done

L. 237: It's a bit of a stretch to write that skull shape is influenced by rural and urban habitat. I would recommend writing something along the line of: "skull shape is different in foxes urban vs. rural foxes".

Rephrased as 'different' line 282

Figure 2: Would it be possible to choose the colours so that they can also be discerned in black and white? Also, the numbers on the axes are very small and I would recommend to make them bigger.

done

Figure 3 and 4: I would recommend to NOT magnify the shape differences. This is misleading. Especially because you write on L. 242 that you found "extensive" differences. I doubt that these differences are that extensive if they are not magnified (but I would be happy to be proven wrong).

We respectfully disagree with this suggestion as it does not mislead the audience, it makes it easier for the audience to discern the anatomical variation. It is also common practice to magnify such variation in cases of microevolution. To us the differences are not large but are in fact 'extensive' in that they occur over the entire skull, and the magnification allows for this to be easily seen.

L. 249 & 250: Do you mean temporalis and masseter "muscle"? Please complement.

Edited on lines 296-298

L. 261: Do you mean increased jaw "closing" speed? Please specify.

We unsure of the issue, but we have tried to clarify this text, line 304, 309. Increased speed means the jaw closes faster, reductions in speed should be self-explanatory

L. 263.264: Isn't the olfactory sense also important in rural areas? Please explain why you think it is more important in urban areas.

We have edited 'should' to 'could' to reflect this important point.

L. 270 ff.: You found that in an interspecific comparison, body size differences influence skull shape in Vulpini. Could the shape differences seen in male and female red foxes also be related to size? If yes, I would be careful to argue with adaptive arguments here. Could you elaborate on this please?

We are unsure but presume that the reviewer is suggesting that allometric variation could be contributing to the shape differences seen between sexes. This is an intriguing question and indeed possible because we chose to simply use sexually mature individuals rather than control for allometry. However, this comment also seems to suggest that allometric variation is generally not adaptive? We would disagree with this view if so.

We took this approach because we felt statistically limiting the effects of allometry could remove biologically important, even functional variation, and decrease our ability to detect differences. Indeed, foxes have determinant growth which is largely complete by sexual maturation. Specific shape/size relationships (i.e. allometries) could be favoured by selection and lead to adaptation.

We feel that being transparent about the inclusion of allometric effects is likely the best policy for our manuscript. There's a paper on such considerations - Outomuro, D, Johansson, F. A potential pitfall in studies of biological shape: Does size matter?. J Anim Ecol. 2017; 86: 1447– 1457.

L. 284: Are these the results corrected for influence of differences in body size? Or the raw data? Please specify, as this is important for the interpretation (especially because PC1 in non-size corrected data is usually strongly correlated with body size)

Yes, we felt this was potentially a more pressing issue at the interspecific level. These steps were specified originally on lines 177-182 in the previous version

And again, it is not just the genus *Vulpes*, but the clade *Vulpini*.

Done

Figure 5: *O. megalotis* and *N. procyonides* are not ancestors of the *Vulpes* species! This is as incorrect as if one would say: Chips are ancestors of humans. They are "basal" or "sister taxa". The ancestor of *Vulpes* is probably extinct. Please rephrase.

We now refer to them as basal

Please report the percentage of variation for each PC on the axis.

We now provide these on the figure.

L. 288-289: I disagree. On PC1, *V. vulpes* is actually the species which is the most (or second most) negative.

Correct, within *Vulpes* proper it is also the most negative, and closest to the basal species. We now discuss this insight on lines 346-350

Further, you do not discuss PC2. How do you interpret these results?

We have now inserted this into the text on lines 346-350.

L. 330-332: "Urban tameness" is indeed a termed – as you surely know – and I would recommend to cite e.g. the following article or a related one:

Uchida, K., Suzuki, K. K., Shimamoto, T., Yanagawa, H., & Koizumi, I. (2019). Decreased vigilance or habituation to humans? Mechanisms on increased boldness in urban animals. *Behavioral Ecology*, 30(6), 1583-1590.

Excellent reference for backing up our interpretation! We now cite this.

L. 342: You are discussing adaptation to an urban environment. But what about non-adaptive processes? For example, the urban population might be characterised by a small founder population (the few individuals which are able to cope with the urban setting), which is again associated with genetic drift. I think that the possibility of the observed patterns being the

result of random processes should be discussed. Further, I think it would be worthwhile mentioning the effects of "ecological release" on skull shape: urban foxes do not have to hunt as frequently as their rural conspecifics, which reduces selection pressures, which in turn may increase variability of skull shape. These are all non-adaptive effects which should be discussed.

We now address these important ideas on lines 393, 406-410 and think they are a direction for future research.

Referee: 3

Comments to the Author(s).

I'm afraid that the statistics is still gobbledeygook to me. This may be my unfamiliarity with the methods, but the editor has to ask how widely the paper should be comprehensible. The idea of showing that the urban-rural effect is in line with the generic line seems very sensible, and I have no problem with that. But what is said in the MS about the machinery employed makes no sense to me.

The statistical approaches are widely used methods, widely available in the R geomorph package, or the IMP morphometrics package and have been cited hundreds, if not thousands of times. However, for starters we have published using the same techniques several times. This includes comparisons of trajectories using confidence intervals in the following papers, which seems to be a focus of the comment by this reviewer:

KJ Parsons, M Concannon, D Navon, et al. 2016. *Molecular ecology* 25, 6012-6023 (cited 42 times)
WJ Cooper, K Parsons, A McIntyre, et al. 2010. *PLoS One* 5: e9551 (cited 118 times)
KJ Parsons, HD Sheets, S Skúlason, MM Ferguson 2011. *Journal of evolutionary biology* 24 (8), 1640-1652 (cited 50 times)

We think the citation rates of these papers, which are actually more technically complex than the current manuscript, speak to their appeal and understanding by the broader community.

First, I return to the question of confidence intervals. The authors disagree with me in their response to referees. Specifically, they write "We disagree that we have misinterpreted our analysis based on confidence intervals. The width of a confidence interval can indeed be due to inadequacy of data as the reviewer asserts, but it can also be due to real variation in the data." To return to basics, an X% confidence interval is defined for a parameter of a statistical model as the range of values that cannot be rejected at the X% level. My difficulty with the use of confidence intervals in the MS include (i) the angle is not a parameter of the model, but something measured after the fit (ii) a confidence interval becomes narrower and narrower (approaching zero width) as the amount of data increases. It is (as its definition states) an assertion about our uncertainty about a parameter. I do not see how a comparison between the width of two confidence intervals in two different datasets can reflect on anything other than our relative uncertainty about the parameters on the basis of the

information in one model versus the other (iii) the confidence intervals themselves seem not to be given, as I read lines 307--308. I can only understand this is saying that (for example) the width of the confidence interval for dorsal-rural/urban is 178 degrees, while the width for dorsal-genus-variation confidence interval is 48 degrees. The confidence interval itself should be a range from one angle to another, and it seems odd not to give that range. It may well be that the term confidence interval is used in this area of biology in a nonstandard way -- perhaps others have done so too. But I am perfectly clear that the MS's use of "confidence interval" is incompatible with the standard definition.

Regarding point (i), *there is **no model** as no variable is being used to explain another, nor is anything being fit*. It is **standard** for a bootstrapping approach to use an observed value that has been **measured** (angle between Vulpini, and urban/rural divergence trajectories) and use random resampling to create confidence intervals to determine whether random processes differ from that observed value. This simply is the logic of bootstrapping and it is commonly used where a model doesn't exist or isn't possible, and we refer the reviewer to the textbook literature on this topic as well as statistical modelling in general. We also now cite the page number of the book where descriptions of the processes in *vecompare* occur on line 264.

Regarding point (ii), we use confidence intervals correctly and in the **standard** way. The reviewer states 'I do not see how a comparison between the width of two confidence intervals in two different datasets can reflect on anything other than our relative uncertainty about the parameters on the basis of the information in one model versus the other'. Again, there is **no model** and these are not two different data sets, they are from a single data set with groups, and all individuals have been measured on a common scale. Also, when we use two confidence intervals (see our new test of sexual dimorphism), we **do not** compare widths of confidence intervals, instead our test determines whether the **confidence intervals of the groups overlap**. This is central to how such bootstrapping tests work and we refer the reviewer to the literature on this topic. However, beyond the literature, this approach for comparing trajectories (which relies on a **standard** bootstrapping approach) has been used for some years in morphometrics, and has been implemented by the leading experts in the field in the major software packages including the R *geomorph* package (Dean Adams et al.), the Matlab *IMP* package (David Sheets), and the java *MorphoJ* package (Christian Klingenberg). The core methodology which follows this **same** use of confidence intervals has been published for some time (Adams and Collyer 2009, *Evolution*) and has been cited over 200 times. The R package implementing this approach has been cited 1044 times (Adams 2013), the book accompanying the *IMP* packages including *vecompare* (as used here) and many other CI-based tests has been cited 3689 times (Zelditch et al. 2004), the *MorphoJ* package, also implementing confidence intervals in the **same** way, has been cited 2551 times. Given the ubiquity of these packages in the field of morphometrics, along with thousands of other published papers utilizing methods derived from them, we are in fact following the 'standard definition' of confidence intervals in the current manuscript.

Regarding point iii, we previously provided the confidence intervals in the text, these remain with some rewording on line 363-364.

Second, my questions about estimation of K have stimulated a new power analysis, mentioned around line 205 and provided in lines 293-296. The MS states that their analysis exhibits "high power to detect differences among models", and gives significant p-values. But it doesn't say which models were being compared, and so the statement provides little information. Generally, a power analysis would have to specify the magnitude of the deviation from the null hypothesis that was being tested. That is, one might say a test has only 30% power against a specified small deviation, but 90% power against some specified larger deviation. The MS quotes power levels, but without stating the parameter values (presumably for K) used to provide them. The "critical test statistic" mentioned on lines 295-296 is pointless to give, as it is not at all explained. Thus, the power analyses need, at least, more explanation.

Details on the power analysis details are expanded and clarified in the methods.

Third, there is the question of supporting null hypotheses. The authors now claim to provide support for a null hypothesis on line 309. They also affirm a null hypothesis ("indicating both views lack a phylogenetic signal") on line 292. The logic of hypothesis testing simply doesn't provide for supporting or affirming a null hypothesis. The type of statement presumably desired would be statistically justifiably provided by a confidence interval for some parameter, showing that the CI is narrow around some particular value. Of course, that relies on a judgment of narrowness, which the author will make, but which the reader will be able to judge for himself or herself. Whether a mean ± 0.1 is close enough to the particular value or not to matter, rather depends on the context, and on the question at hand. But there is no sense in using the term "null hypothesis" for the particular value.

It is standard scientific method to propose a null and alternate hypothesis. Previously we noted that the null hypothesis was supported and explicitly added this phrasing to clarify a recent round of reviews. Simply put, our methods tested the alternative hypothesis which was not supported. According to the scientific method, this simply means that the null is supported. For trajectories we have done two separate tests. Our null is statistically supported, with values being 'narrow' around a particular value. This is even 'convincing' the reviewer that trajectory is shared as they state in their final comment so we are confused about what the issue is.

Finally, I didn't follow fully all the stuff about regressing landmarks on canonical variables, but I think I get the gist. (I presume that canonical variables were produced from the landmark data.) My question here is: in order to show that the direction of change in urban/rural is the same as within the genus, why is one not comparing the canonical variables?

The reviewer appears to be mistaken about the makeup of, or limitations of, comparing canonical variables directly. Firstly, there are only canonical variables for the urban/rural

divergence and not for Vulpini, so this makes such an approach impossible. However, in line with what we think the reviewer is suggesting we are in fact comparing **the effect** of variables that describe urban/rural and clade level divergence on the **trajectory** of shape variation.

In short, it is not possible to compare the direction of change by simply comparing Vulpini PC1 and urban/rural 'canonical variables', because they differ in scale (this is because different models are needed to create them). The test we use, which is common in the field, relies on trigonometry (i.e. the arc cosine is used to calculate the angle of change). This, simply by the nature of trigonometry, requires a shape which we have in the form of landmarks. It is *the only* valid way we know of that this type of comparison can be done. Therefore, for this and several other reasons plain/straight-up comparisons of canonical variables can't do anything to inform us about **the direction** of shape change.

In particular, the regression seems to be asking how the urban/rural data differ from *their* canonical axis, and how the inter-specific data differ from *their* canonical axis. Why are those deviations within each dataset relevant to whether the canonical variables are the same? I can see that the deviations within a dataset could be used to estimate the certainty of the canonical variable loadings, and that this could in principle lead to a test of whether the two canonical variables differ. BUT, comparing the angles of the intra-dataset comparisons doesn't take into account the difference between the canonical variables, which surely must be relevant.

The reviewer is not providing line numbers so we are unsure what this comment is directed at as there is more than one regression analysis in this manuscript. We can assume they are again commenting on the vecompare analysis, but it could also be our analysis in the RRPP package. Either way, both analyses test whether the **trajectory** of canonical variables from the urban/rural **differs** from the primary axis of interspecific variation (i.e. Vulpini variation). This essentially tests whether **patterns** of microevolution match **patterns** of macroevolution. This is a core question in evolution as it is often asserted that macroevolution reflects extended microevolution.

The analysis we perform has nothing to do with concerns or interest in canonical loadings, and it is unclear to us why such canonical loadings would matter to any of the biological questions we aimed to address in the manuscript. We do think that differences between canonical variables are relevant and that is why **we appropriately tested their effect on shape**, and did so in two different ways that are standard in the field of morphometrics.

Further, we have previously explained our methods to a level of detail that is standard within the field, (see Parsons et al. 2018, Proc. B 1877, 20172671), and it is beyond the scope of our manuscript, and page limits in Proc B to explain every facet of the analysis. We are using common methods, none are bespoke, multivariate statistics are almost always needed to analyse shape data, so if the reviewer or audience is unsure of how the methods work we have provided references to the core methodological publications that have the scope to provide every relevant detail.

The authors may feel that they have important anatomical points to make about fox skulls and domestication, and that the statistics is really unimportant. It may well be that a much simpler statistical argument would be fully acceptable to justify their biological conclusions. I have no reason to doubt their biological conclusions that the urban/rural changes are in line with the inter-specific changes. But I hope I've explained why the statistical aspect of the paper has a number of problematic elements.

We do indeed have important points to make about foxes, and using the **appropriate** stats is vital to getting the story correct. The statistics we use are not conventional (i.e. t-tests, linear regression, ANOVAs), but are standard in the field because they are necessary for the type of data we have.

The reviewer suggests 'it may well be that a much simpler statistical argument' could be used, but provides no suggestions for what these simpler alternatives could be. We are confident that there are no problematic elements, that there are very limited statistical alternatives, and that 'simple' approaches would not enable us to address our questions.

Some minor points:

214. Drop the comma? (Or indeed move earlier in sentence)

done

219. Is their approach scale free? PCA isn't!

We use the trajectory of PC1 in our analysis, which is scale free.

312. Should it be $p < 0.001$?? rather than >

changed

316 "way to divergence" : the "to" reads very oddly -- as?

changed, line 378

Reference 38 is a non-functioning URL. I was hoping to find some explanation of their calculations with regress8.

The host site recently changed, and we now provide and cite the latest link

<https://www.animal-behaviour.de/imp/>

Appendix C

Associate Editor Board Member: 1

Comments to Author:

As mentioned previously, I think this paper deals with topical subject, and the authors have accumulated a very nice anatomical data set with which to address the questions surrounding urbanisation on animal form-function. The reviewers generally share this positive outlook on both the topic and the attempt to analyse the data. However, reviewer 2 suggests some reasonable improvements to improve clarity and smooth out the rough edges on certain interpretations. Reviewer 3 remains concerned about some of the statistical analyses and particularly the way they are rationalised and presented. It is my recommendation that the authors be given the opportunity to resubmit a modified version to incorporate the changes suggested by Reviewers 2 & 3.

Dear Editor,

We thank you for these latest reviews and have responded to them below. We are pleased that reviewer 1 sees no further need for change, and that reviewer 4 has provided a number of helpful comments and insights that will improve the manuscript. We have taken most of these points on board as you will see below. However, we are unclear why reviewer 3 is questioning our statistical methods, given that these are widely used and for which we have provided numerous references. We respond below to reviewer 3's comments as best we can but do not feel it is necessary to change our analyses given their widespread usage.

Reviewer(s)' Comments to Author:

Referee: 1

Comments to the Author(s).

I am satisfied that the authors have addressed all my comments appropriately. As far as I can tell they have also addressed the comments of the other reviewers in a sensible fashion.

We thank the reviewer for this opinion

Referee: 4

Comments to the Author(s).

This manuscript is about cranial shape variation in urban vs. rural red foxes. The authors found that there are significant differences in skull shape between urban and rural foxes and that these shape changes are consistent with the ones found in an interspecific comparison among other *Vulpini*. This is an interesting study on the subject of urban adaptation, which is still far to less extensively investigated, despite its rising importance in a steadily urbanizing world.

The data set is exceptionally extensive and, as far as I can judge, the applied methods are sound. However, I feel that the manuscript lacks clarity in many instances (see my detailed comments below). The terminology and the descriptions related to taxonomy, systematics, and concepts of evolution, development, domestication, and breed formation are oftentimes quite peculiar and unusual compared to subject-specific literature, as well as too simplified, in some parts at the brink to non-correct (again, see my detailed comments below). Further, I would strongly recommend to include further references in various instances (also as indicated below) and to extend the discussion. For example, the authors discuss only adaptive concepts, without taking into account non-adaptive processes, which might explain their findings.

We thank the reviewer for this constructive overview. We feel we have an especially powerful dataset that enables us to address a broad set of topics. Most of the comments provided here have now been taken on board unless stated otherwise. Several references have been added and sections clarified as suggested.

Detailed suggestions for edits:

L. 15: Genus name *Vulpes* in upper case.

Changed throughout

L. 20-21: I think your sampling of the genus *Vulpes* is actually not complete, as stated here. According to the standard reference work by Wilson & Reeder (Mammal species of the world, 2005, 3rd edition), there is also *Vulpes pallida*, which you did not sample. Please rephrase, or specify later which reference for *Vulpes* you were using.

Thanks, we have rephrased this by replacing 'complete' with 'extensive' which is more accurate (line 21)

L. 23 & L. 40: Do you mean divergence "of skull shape" between urban and rural habitats"? Please make clear what you mean. Otherwise it sounds as if the habitats themselves would diverge.

Thanks, we have changed these lines to clarify that we mean phenotypes on line 41

L. 24-25: This sentence basically repeats the previous one. Maybe delete?

We have now edited this sentence.

L. 25-26: Why "additionally"? Dogs are one example of many where this shortening of the

snout has occurred during domestication. Plus, please be aware that domestication is not equal to breed formation, so the term “breed” should not be used here.

We have removed dogs as an example. We do however think they are of particular interest in relation to foxes because they are both canids. Therefore, we feel that our data set could lend itself especially well toward hypotheses explaining the initial steps in the domestication of wolves. Line 27

L. 37-40: Maybe add here, that selection could not only be different across a suite of functionally salient traits, but also across clades.

We are unsure what the reviewer means here by clades. However, in relation to this we have added in the mention that we do not know whether this divergence is present in replicate sites (i.e. other cities and adjacent rural sites). Line 44

L. 48-49: It would be appropriate to cite also the original paper by the people who came up with the term ‘domestication syndrome’: Wilkins et al. 2014 (I think it’s your reference 44).

Thanks, we have cited this now.

L. 48 & 51: What do you mean by “stereotypical”? That similar changes occur in different domestic species? Please clarify.

We now specify different species

L. 50-53: This list is not exhaustive and I would recommend to include “for example” or something similar.

Done

Paragraph on domestication syndrome: I would strongly recommend to also cite the new paper by Lord et al. 2019, who critically discuss the ‘domestication syndrome’: Lord, K. A., Larson, G., Coppinger, R. P., & Karlsson, E. K. (2019). The history of farm foxes undermines the animal domestication syndrome. Trends in Ecology & Evolution.

Very interesting paper! We now cite this, and it is an important caveat that some amount of domestication in this famous experiment could have been pre-dated by the source of foxes used. Line 80

Further, and I have already mentioned this above: the selection for a shorter snout in some dog breeds (as described e.g. by Drake & Klingenberg 2010) has nothing to do with the domestication syndrome, but with breed formation. This means, with selection for specific traits subsequent to domestication. Domestication and breed formation are two very different processes and should not be mixed.

Fine comment, we have clarified this distinction in the text. However, we do respectfully disagree that they have nothing to do with each other as developmental bias may unify both processes.

L. 55: There is no evidence that the extensive morphological change in dogs has been particularly fast. See e.g. Geiger, M., & Sánchez-Villagra, M. R. (2018). Similar rates of morphological evolution in domesticated and wild pigs and dogs. *Frontiers in zoology*, 15(1), 23.

We remove the use of 'rapid' here, but it does seem that both lineages (pigs and dogs) have undergone rapid change relative to most mammals we are aware of.

I would further strongly recommend to refer to the domestic form of dogs as 'domestic dogs' or the like, not just 'dogs'. This could be confused with you referring to the clade Canidae, which is usually also referred to as 'dogs'.

We have changed this throughout the text

L.58: Developmental biases 'within' dog skulls is a phrasing I've never heard before. I would delete this.

Additionally: I think artificial selection for these traits and developmental biases are not mutually exclusive. It may be that head shape development is biased in one way or the other, leading to a limited number of possible morphotypes within an evolutionary lineage. However, such morphotypes, e.g. as seen in bulldog or greyhound, would not be prevalent in these breeds if it wasn't for artificial selection, i.e., people who were selecting for these traits as they occurred.

We have removed 'within'. We agree with this opinion and it was our intention to get this point across in the text. Therefore, we have edited to enhance this point.

L. 61: Developmental modularity does not 'favour' changes in the snout independent of the rest of the skull, but it 'makes possible' (or something like this). These are developmental patterns, not "conscious" processes.

Agreed and changed to reflect this.

L. 65-66: '...shows a similar trend in skull shape.' What exactly? Skull shape change? Please specify.

We have collapsed the last two sentences in the paragraph to accommodate this point

L. 67: What do you mean by "mimic ancestral condition for dogs?" It is not clear which condition you mean. I would suggest to put write something in the line of "mimic the selection regime during the initial domestication process of dogs".

Changed in accordance with this suggestion on line 94

You could also cite here this study on mice, in which selection for tameness was more similar to the scenario in dogs and other commensal domesticates (and has led to a shortening of the head): Geiger, M., Sánchez-Villagra, M. R., & Lindholm, A. K. (2018). A longitudinal study of phenotypic changes in early domestication of house mice. *Royal Society open science*, 5(3), 172099.

Interesting paper, we are focused on canids in this section so felt it was a bit awkward to include this without adding substantial length (we are already pushing the limits for Proc b)

L. 69: "during" instead of "for"?

Done

L. 70: "driver of initial changes" of what? Please specify.

Toward domestication now added

L. 78: Why do you point out that foxes are non-social? Does this have an implication for this study? In fact, I would not regard foxes as non-social per-se, as they are sometimes living in family groups and are showing complex social behaviour. I agree that the behaviour may be less complex than in wolves (e.g., solitary vs. pack hunting), but still not non-social.

We concede that foxes have some level of social behaviour. Foxes live in monogamous pairs but not social groups such as feral dogs and wolves. It is our view that if social behaviour is seen as a driver of domestication in dogs, it is also confounded with other phenotypes. With foxes we see a much less social animal, yet a similar trajectory of morphological change (i.e. morphology is no longer confounded from the exceptionally social behaviours of dogs).

L. 84: City foxes are not only known from Great Britain, but also from Central Europe: Gloor, S., Bontadina, F., Hegglin, D., & Deplazes 2001. The rise of urban fox populations in Switzerland. *Mammalian Biology*, 66, 155-164.

Thanks, yes that is noteworthy. We cite this a bit later in the manuscript now on line 104

L. 89: In this study on urban and rural foxes in Switzerland, it has been shown that urban foxes are indeed somewhat isolated from their rural conspecifics: Wandeler, P., Funk, S. M., Lurgiader, C. R., Gloor, S., & Breitenmoser, U. (2003). The city-fox phenomenon: Genetic consequences of a recent colonization of urban habitat. *Molecular Ecology*, 12(3), 647-656.

Very useful reference, we have included this now on line 104

L.94ff: You predict that rostral shape in urban foxes would follow the line of the domestication syndrome, but for brain size and sexual dimorphism it would be the opposite. This is contradictory. After all, in urban house finches, it was found that brain and eye size is smaller compared to rural conspecifics (Hutton & McGraw 2016). Further, in general, domestication is associated with a decrease of sexual dimorphism.

Hutton, P., & McGraw, K. J. (2016). Urban–rural differences in eye, bill, and skull allometry in house finches (*Haemorrhous mexicanus*). *Integrative and comparative biology*, 56(6), 1215-1224.

Thanks for this input, we now point out that our brain size prediction contradicts domestication syndrome (line 113) but to remain objective we leave the sexual dimorphism hypothesis as is (even though shape differences indicate it may be smaller in the urban habitat).

Our sexual dimorphism hypothesis was directed at differential effects of habitat on sexes, but was not directional whereby we would expect sexual dimorphism to decrease in the urban habitat. So, this raised an interesting question and we have now performed an additional analysis to address this directional prediction. Interestingly we now find support for this prediction where the ventral aspect trends toward reduced dimorphism in the urban habitat, and the dorsal aspect shows a significantly reduced amount of dimorphism. The methods for this are now provided, and findings are now briefly discussed. We thank the reviewer for these ideas as it strengthens support for urban habitats matching up with domestication syndrome. Methods described in Lines 177-187

Hutton and McGraw is cited on line 51 now to set up this prediction

L. 100-102: The comparison rural vs. urban is intraspecific. I do not see why these should be biased by phylogenetic relatedness and why this analysis is important here.

We feel the reviewer has misinterpreted our intentions, or misunderstands our interests. We certainly would not expect phylogenetic effects to differ between urban and rural populations. However, we did feel that phylogenetic effects within the Vulpini could be influencing the response to urban/rural habitat gradients. We have clarified this text to get the point across more explicitly. While we did not find phylogenetic effects on shape difference within Vulpini, the alignment of their divergence with urban/rural divergence is suggestive of developmental bias influencing evolution at different scales. Lines 121-125

L. 108 ff. (Materials and Methods): Would it be possible to provide the number of specimens in each habitat (rural and urban) as well as the sex in a table or written out in the text in the main manuscript, please? This would be crucial for the interpretation of the results, especially to see if there were e.g. equal numbers of males and females for each habitat. It's good to have the supplementary table with all the specimens, but a summary would also be important for the main text.

Done, although we have done this in the text to save the space needed for another table. Line 131-133

L. 111: I would recommend to write: Which are "housed" in the collections of the...

done, line 133

L. 113: All specimens "have"
done, line 135

L. 124-125: So I guess the fusion of the basisphenoid synchondrosis is associate with age?
Could you please provide a reference here?

Done, line 147 with a new reference provided:

Roulichova J, Jana AM. 2007. Simple method of age determination in red fox, *Vulpes vulpes*.
Folia Zoologica 56, 440-444

L. 129: Maybe "aspects" instead of "views"?

Changed throughout

L. 129-131: This is a peculiar phrasing. Dorsal and ventral aspects do not "provide" landmarks.
Rather, you have defined these landmarks on these aspects. Please rephrase.

Some minor rephrasing has fixed this on lines 151-153

Figure 1:

- Delete "the" and "genus" in "the *Vulpes* genus". It's clear that *Vulpes* is a genus.
- Landmark descriptions are actually in supplementary table 3.

Done but it is supplementary table 2 where the landmarks are

Supplementary Table 3.

- Landmark 5: delete duplication of words.
- What is the difference between landmarks 2 and 3 (and 7 and 8)? Please describe 2 and 7 more accurately.

Supplementary table 2 actually, landmark 3 is at the *maxillary* suture, landmark 2 is at the anterior *tip of maxillary* suture. Landmark 5 is fixed for repetition.

L. 135: The last part of this sentence seems not to belong there. Rephrase?
done

L. 136: This is actually Supp. Table 2, not 1. I think this table, or at least the names of the species, should go in the main text.

We are unsure of what the error is here, the species sampling table is supplementary table 3 in our submission. It should remain supp. Table 3 because that is the order of topics and

where the text requires its use.

Further, what do you mean by "additional canids"? As far as I see, you have sampled almost the whole clade *Vulpes* (for the one exception see my comment above) as well as the basal *Otocyon* and *Nyctereutes*. This means you have basically sampled the clade *Vulpini*. Please make this clear, as this phylogenetic context is important to understand the implications of your analysis.

We have clarified this text and added in the species list. Lines 155,158-161

L. 162: what do you mean by "they"? Do you mean you want to test if similar shape changes as seen in interspecific comparison are also prevalent in an intraspecific (urban vs. rural) comparison?

This sentence has now been clarified, line 198

L. 165 ff.: which one is the dependent and which the independent variable? Please specify. Please write again what you mean by "shape". Partial warp scores?

We stated previously in our text that partial warp scores accounted for quantitative variation in shape, line 166 . We also stated that sex, habitat, and their interaction were explanatory variables, while shape was the response variable on lines 175,176.

Also, I do not see why you performed an ANOVA first. Aren't both of your variables continuous? You were performing a regression analysis anyway, so this tells you the same thing. Or do I miss something here?

We did not perform a normal ANOVA anywhere, but we did do a Procrustes. A standard regression approach is not appropriate to statistically test the effect of categorical explanatory variables like habitat and sex. We later use a regression to model the effect of these variables on shape to produce visual aids in the form of deformation grids (line 204), but we do not use it to test the effect of these factors.

Please rephrase: there is no such a thing as "effect of allometry on shape". It would be "effect of size on shape", which is allometry or allometric scaling.

We have edited this on line 201

L. 170: You say you did this for *Vulpes* species, however, in the Figure showing the results of PCA, there are also the other two non-*Vulpes* species of *Vulpini*. Please rephrase.

Rephrased to account for the two basal species. Line 208

L. 171: this is actually supplementary table 4, not 3.

Need to check

L. 181: It is not 12 Vulpes species (see above), but 12 species of Vulpini. Please change (also in remainder of the text).

Rephrased throughout

L. 197: Remove brackets around (BM). I would suggest to not abbreviate this term anyway. Every abbreviation makes a text harder to read and this specific term only occurs three times in the manuscript.

Changed throughout

L.161-161 and L.184-185: These two paragraphs have a similar introductory sentence. This is confusing and I would suggest to condensate.

We have edited these so they are more easily distinguished now

L. 209: You did not perform a 'phylogenetic analysis'. This would be the analysis of phylogenetic relationships between clades. Please rephrase.

We have made a minor adjustment of the text on line 250

L. 208-211: This is confusing. You are performing the test to determine if phylogeny has an effect, but saying here that there is no need to account for a phylogenetic signal. Maybe I do not get the point here, but I think it would be helpful to rephrase.

We have approached our data objectively but have also now edited this text in the manuscript to address these concerns on lines 249

L. 224: delete one "run"

Done

L. 237: It's a bit of a stretch to write that skull shape is influenced by rural and urban habitat. I would recommend writing something along the line of: "skull shape is different in foxes urban vs. rural foxes".

Rephrased as 'different' line 282

Figure 2: Would it be possible to choose the colours so that they can also be discerned in black and white? Also, the numbers on the axes are very small and I would recommend to make them bigger.

done

Figure 3 and 4: I would recommend to NOT magnify the shape differences. This is misleading. Especially because you write on L. 242 that you found "extensive" differences. I doubt that these differences are that extensive if they are not magnified (but I would be happy to be proven wrong).

We respectfully disagree with this suggestion as it does not mislead the audience, it makes it easier for the audience to discern the anatomical variation. It is also common practice to magnify such variation in cases of microevolution. To us the differences are not large but are in fact 'extensive' in that they occur over the entire skull, and the magnification allows for this to be easily seen.

L. 249 & 250: Do you mean temporalis and masseter "muscle"? Please complement.

Edited on lines 296-298

L. 261: Do you mean increased jaw "closing" speed? Please specify.

We unsure of the issue, but we have tried to clarify this text, line 304, 309. Increased speed means the jaw closes faster, reductions in speed should be self-explanatory

L. 263.264: Isn't the olfactory sense also important in rural areas? Please explain why you think it is more important in urban areas.

We have edited 'should' to 'could' to reflect this important point.

L. 270 ff.: You found that in an interspecific comparison, body size differences influence skull shape in Vulpini. Could the shape differences seen in male and female red foxes also be related to size? If yes, I would be careful to argue with adaptive arguments here. Could you elaborate on this please?

We are unsure but presume that the reviewer is suggesting that allometric variation could be contributing to the shape differences seen between sexes. This is an intriguing question and indeed possible because we chose to simply use sexually mature individuals rather than control for allometry. However, this comment also seems to suggest that allometric variation is generally not adaptive? We would disagree with this view if so.

We took this approach because we felt statistically limiting the effects of allometry could remove biologically important, even functional variation, and decrease our ability to detect differences. Indeed, foxes have determinant growth which is largely complete by sexual maturation. Specific shape/size relationships (i.e. allometries) could be favoured by selection and lead to adaptation.

We feel that being transparent about the inclusion of allometric effects is likely the best policy for our manuscript. There's a paper on such considerations - Outomuro, D, Johansson, F. A potential pitfall in studies of biological shape: Does size matter?. J Anim Ecol. 2017; 86: 1447– 1457.

L. 284: Are these the results corrected for influence of differences in body size? Or the raw data? Please specify, as this is important for the interpretation (especially because PC1 in non-size corrected data is usually strongly correlated with body size)

Yes, we felt this was potentially a more pressing issue at the interspecific level. These steps were specified originally on lines 177-182 in the previous version

And again, it is not just the genus *Vulpes*, but the clade *Vulpini*.

Done

Figure 5: *O. megalotis* and *N. procyonides* are not ancestors of the *Vulpes* species! This is as incorrect as if one would say: Chips are ancestors of humans. They are "basal" or "sister taxa". The ancestor of *Vulpes* is probably extinct. Please rephrase.

We now refer to them as basal

Please report the percentage of variation for each PC on the axis.

We now provide these on the figure.

L. 288-289: I disagree. On PC1, *V. vulpes* is actually the species which is the most (or second most) negative.

Correct, within *Vulpes* proper it is also the most negative, and closest to the basal species. We now discuss this insight on lines 346-350

Further, you do not discuss PC2. How do you interpret these results?

We have now inserted this into the text on lines 346-350.

L. 330-332: "Urban tameness" is indeed a termed – as you surely know – and I would recommend to cite e.g. the following article or a related one:

Uchida, K., Suzuki, K. K., Shimamoto, T., Yanagawa, H., & Koizumi, I. (2019). Decreased vigilance or habituation to humans? Mechanisms on increased boldness in urban animals. *Behavioral Ecology*, 30(6), 1583-1590.

Excellent reference for backing up our interpretation! We now cite this.

L. 342: You are discussing adaptation to an urban environment. But what about non-adaptive processes? For example, the urban population might be characterised by a small founder population (the few individuals which are able to cope with the urban setting), which is again associated with genetic drift. I think that the possibility of the observed patterns being the

result of random processes should be discussed. Further, I think it would be worthwhile mentioning the effects of "ecological release" on skull shape: urban foxes do not have to hunt as frequently as their rural conspecifics, which reduces selection pressures, which in turn may increase variability of skull shape. These are all non-adaptive effects which should be discussed.

We now address these important ideas on lines 393, 406-410 and think they are a direction for future research.

Referee: 3

Comments to the Author(s).

I'm afraid that the statistics is still gobbledeygook to me. This may be my unfamiliarity with the methods, but the editor has to ask how widely the paper should be comprehensible. The idea of showing that the urban-rural effect is in line with the generic line seems very sensible, and I have no problem with that. But what is said in the MS about the machinery employed makes no sense to me.

The statistical approaches are widely used methods, widely available in the R geomorph package, or the IMP morphometrics package and have been cited hundreds, if not thousands of times. However, for starters we have published using the same techniques several times. This includes comparisons of trajectories using confidence intervals in the following papers, which seems to be a focus of the comment by this reviewer:

KJ Parsons, M Concannon, D Navon, et al. 2016. *Molecular ecology* 25, 6012-6023 (cited 42 times)
WJ Cooper, K Parsons, A McIntyre, et al. 2010. *PLoS One* 5: e9551 (cited 118 times)
KJ Parsons, HD Sheets, S Skúlason, MM Ferguson 2011. *Journal of evolutionary biology* 24 (8), 1640-1652 (cited 50 times)

We think the citation rates of these papers, which are actually more technically complex than the current manuscript, speak to their appeal and understanding by the broader community.

First, I return to the question of confidence intervals. The authors disagree with me in their response to referees. Specifically, they write "We disagree that we have misinterpreted our analysis based on confidence intervals. The width of a confidence interval can indeed be due to inadequacy of data as the reviewer asserts, but it can also be due to real variation in the data." To return to basics, an X% confidence interval is defined for a parameter of a statistical model as the range of values that cannot be rejected at the X% level. My difficulty with the use of confidence intervals in the MS include (i) the angle is not a parameter of the model, but something measured after the fit (ii) a confidence interval becomes narrower and narrower (approaching zero width) as the amount of data increases. It is (as its definition states) an assertion about our uncertainty about a parameter. I do not see how a comparison between the width of two confidence intervals in two different datasets can reflect on anything other than our relative uncertainty about the parameters on the basis of the

information in one model versus the other (iii) the confidence intervals themselves seem not to be given, as I read lines 307--308. I can only understand this is saying that (for example) the width of the confidence interval for dorsal-rural/urban is 178 degrees, while the width for dorsal-genus-variation confidence interval is 48 degrees. The confidence interval itself should be a range from one angle to another, and it seems odd not to give that range. It may well be that the term confidence interval is used in this area of biology in a nonstandard way -- perhaps others have done so too. But I am perfectly clear that the MS's use of "confidence interval" is incompatible with the standard definition.

Regarding point (i), *there is **no model** as no variable is being used to explain another, nor is anything being fit*. It is **standard** for a bootstrapping approach to use an observed value that has been **measured** (angle between Vulpini, and urban/rural divergence trajectories) and use random resampling to create confidence intervals to determine whether random processes differ from that observed value. This simply is the logic of bootstrapping and it is commonly used where a model doesn't exist or isn't possible, and we refer the reviewer to the textbook literature on this topic as well as statistical modelling in general. We also now cite the page number of the book where descriptions of the processes in *vecompare* occur on line 264.

Regarding point (ii), we use confidence intervals correctly and in the **standard** way. The reviewer states 'I do not see how a comparison between the width of two confidence intervals in two different datasets can reflect on anything other than our relative uncertainty about the parameters on the basis of the information in one model versus the other'. Again, there is **no model** and these are not two different data sets, they are from a single data set with groups, and all individuals have been measured on a common scale. Also, when we use two confidence intervals (see our new test of sexual dimorphism), we **do not** compare widths of confidence intervals, instead our test determines whether the **confidence intervals of the groups overlap**. This is central to how such bootstrapping tests work and we refer the reviewer to the literature on this topic. However, beyond the literature, this approach for comparing trajectories (which relies on a **standard** bootstrapping approach) has been used for some years in morphometrics, and has been implemented by the leading experts in the field in the major software packages including the R *geomorph* package (Dean Adams et al.), the Matlab *IMP* package (David Sheets), and the java *MorphoJ* package (Christian Klingenberg). The core methodology which follows this **same** use of confidence intervals has been published for some time (Adams and Collyer 2009, *Evolution*) and has been cited over 200 times. The R package implementing this approach has been cited 1044 times (Adams 2013), the book accompanying the *IMP* packages including *vecompare* (as used here) and many other CI-based tests has been cited 3689 times (Zelditch et al. 2004), the *MorphoJ* package, also implementing confidence intervals in the **same** way, has been cited 2551 times. Given the ubiquity of these packages in the field of morphometrics, along with thousands of other published papers utilizing methods derived from them, we are in fact following the 'standard definition' of confidence intervals in the current manuscript.

Regarding point iii, we previously provided the confidence intervals in the text, these remain with some rewording on line 363-364.

Second, my questions about estimation of K have stimulated a new power analysis, mentioned around line 205 and provided in lines 293-296. The MS states that their analysis exhibits "high power to detect differences among models", and gives significant p-values. But it doesn't say which models were being compared, and so the statement provides little information. Generally, a power analysis would have to specify the magnitude of the deviation from the null hypothesis that was being tested. That is, one might say a test has only 30% power against a specified small deviation, but 90% power against some specified larger deviation. The MS quotes power levels, but without stating the parameter values (presumably for K) used to provide them. The "critical test statistic" mentioned on lines 295-296 is pointless to give, as it is not at all explained. Thus, the power analyses need, at least, more explanation.

Details on the power analysis details are expanded and clarified in the methods.

Third, there is the question of supporting null hypotheses. The authors now claim to provide support for a null hypothesis on line 309. They also affirm a null hypothesis ("indicating both views lack a phylogenetic signal") on line 292. The logic of hypothesis testing simply doesn't provide for supporting or affirming a null hypothesis. The type of statement presumably desired would be statistically justifiably provided by a confidence interval for some parameter, showing that the CI is narrow around some particular value. Of course, that relies on a judgment of narrowness, which the author will make, but which the reader will be able to judge for himself or herself. Whether a mean ± 0.1 is close enough to the particular value or not to matter, rather depends on the context, and on the question at hand. But there is no sense in using the term "null hypothesis" for the particular value.

It is standard scientific method to propose a null and alternate hypothesis. Previously we noted that the null hypothesis was supported and explicitly added this phrasing to clarify a recent round of reviews. Simply put, our methods tested the alternative hypothesis which was not supported. According to the scientific method, this simply means that the null is supported. For trajectories we have done two separate tests. Our null is statistically supported, with values being 'narrow' around a particular value. This is even 'convincing' the reviewer that trajectory is shared as they state in their final comment so we are confused about what the issue is.

Finally, I didn't follow fully all the stuff about regressing landmarks on canonical variables, but I think I get the gist. (I presume that canonical variables were produced from the landmark data.) My question here is: in order to show that the direction of change in urban/rural is the same as within the genus, why is one not comparing the canonical variables?

The reviewer appears to be mistaken about the makeup of, or limitations of, comparing canonical variables directly. Firstly, there are only canonical variables for the urban/rural

divergence and not for Vulpini, so this makes such an approach impossible. However, in line with what we think the reviewer is suggesting we are in fact comparing **the effect** of variables that describe urban/rural and clade level divergence on the **trajectory** of shape variation.

In short, it is not possible to compare the direction of change by simply comparing Vulpini PC1 and urban/rural 'canonical variables', because they differ in scale (this is because different models are needed to create them). The test we use, which is common in the field, relies on trigonometry (i.e. the arc cosine is used to calculate the angle of change). This, simply by the nature of trigonometry, requires a shape which we have in the form of landmarks. It is *the only* valid way we know of that this type of comparison can be done. Therefore, for this and several other reasons plain/straight-up comparisons of canonical variables can't do anything to inform us about **the direction** of shape change.

In particular, the regression seems to be asking how the urban/rural data differ from *their* canonical axis, and how the inter-specific data differ from *their* canonical axis. Why are those deviations within each dataset relevant to whether the canonical variables are the same? I can see that the deviations within a dataset could be used to estimate the certainty of the canonical variable loadings, and that this could in principle lead to a test of whether the two canonical variables differ. BUT, comparing the angles of the intra-dataset comparisons doesn't take into account the difference between the canonical variables, which surely must be relevant.

The reviewer is not providing line numbers so we are unsure what this comment is directed at as there is more than one regression analysis in this manuscript. We can assume they are again commenting on the vecompare analysis, but it could also be our analysis in the RRPP package. Either way, both analyses test whether the **trajectory** of canonical variables from the urban/rural **differs** from the primary axis of interspecific variation (i.e. Vulpini variation). This essentially tests whether **patterns** of microevolution match **patterns** of macroevolution. This is a core question in evolution as it is often asserted that macroevolution reflects extended microevolution.

The analysis we perform has nothing to do with concerns or interest in canonical loadings, and it is unclear to us why such canonical loadings would matter to any of the biological questions we aimed to address in the manuscript. We do think that differences between canonical variables are relevant and that is why **we appropriately tested their effect on shape**, and did so in two different ways that are standard in the field of morphometrics.

Further, we have previously explained our methods to a level of detail that is standard within the field, (see Parsons et al. 2018, Proc. B 1877, 20172671), and it is beyond the scope of our manuscript, and page limits in Proc B to explain every facet of the analysis. We are using common methods, none are bespoke, multivariate statistics are almost always needed to analyse shape data, so if the reviewer or audience is unsure of how the methods work we have provided references to the core methodological publications that have the scope to provide every relevant detail.

The authors may feel that they have important anatomical points to make about fox skulls and domestication, and that the statistics is really unimportant. It may well be that a much simpler statistical argument would be fully acceptable to justify their biological conclusions. I have no reason to doubt their biological conclusions that the urban/rural changes are in line with the inter-specific changes. But I hope I've explained why the statistical aspect of the paper has a number of problematic elements.

We do indeed have important points to make about foxes, and using the **appropriate** stats is vital to getting the story correct. The statistics we use are not conventional (i.e. t-tests, linear regression, ANOVAs), but are standard in the field because they are necessary for the type of data we have.

The reviewer suggests 'it may well be that a much simpler statistical argument' could be used, but provides no suggestions for what these simpler alternatives could be. We are confident that there are no problematic elements, that there are very limited statistical alternatives, and that 'simple' approaches would not enable us to address our questions.

Some minor points:

214. Drop the comma? (Or indeed move earlier in sentence)

done

219. Is their approach scale free? PCA isn't!

We use the trajectory of PC1 in our analysis, which is scale free.

312. Should it be $p < 0.001$?? rather than $>$

changed

316 "way to divergence" : the "to" reads very oddly -- as?

changed, line 378

Reference 38 is a non-functioning URL. I was hoping to find some explanation of their calculations with regress8.

The host site recently changed, and we now provide and cite the latest link

<https://www.animal-behaviour.de/imp/>